# $CCl_4$ distribution derived from MIPAS ESA V7 data: intercomparisons, trend and lifetime estimation

Massimo Valeri[1,2], Flavio Barbara[3], Chris Boone[4], Simone Ceccherini[3], Marco Gai[3], Guido Maucher[6], Piera Raspollini[3], Marco Ridolfi[1,3], Luca Sgheri[5], Gerald Wetzel[6], and Nicola Zoppetti[3]

[1]Dipartimento di Fisica e Astronomia, Università di Bologna, Italy
[2]Istituto di Scienze dell'Atmosfera e del Clima, Consiglio Nazionale delle Ricerche, Bologna, Italy
[3]Istituto di Fisica Applicata "Nello Carrara", Consiglio Nazionale delle Ricerche, Firenze, Italy
[4]Department of Chemistry, University of Waterloo, Waterloo, Ontario, Canada
[5]Istituto per le Applicazioni del Calcolo, Consiglio Nazionale delle Ricerche, Firenze, Italy
[6]Karlsruhe Institute of Technology, Institute of Meteorology and Climate Research, Karlsruhe, Germany

*Correspondence to:* Marco Ridolfi (marco.ridolfi@unibo.it)

**Abstract.** Atmospheric emissions of carbon tetrachloride ($CCl_4$) are regulated by the Montreal Protocol due to its role as a strong ozone-depleting substance. The molecule has been the subject of recent increased interest as a consequence of the so called "mystery of $CCl_4$", the discrepancy between atmospheric observations and reported production and consumption. Surface measurements of $CCl_4$ atmospheric concentrations have declined at a rate almost three times smaller than its lifetime-limited rate, suggesting persistent atmospheric emissions despite the ban. In this paper, we study $CCl_4$ vertical and zonal distributions in the upper troposphere and lower stratosphere (including the photolytic loss region, 70-20 hPa), its trend, and its stratospheric lifetime using measurements from the Michelson Interferometer for Passive Atmospheric Sounding (MIPAS), which operated onboard the ENVISAT satellite from 2002 to 2012. Specifically, we use the MIPAS data product generated with Version 7 of the Level 2 algorithm operated by the European Space Agency.

The $CCl_4$ zonal means show features typical of long-lived species of anthropogenic origin that are destroyed primarily in the stratosphere, with larger quantities in the troposphere and a monotonic decrease with increasing altitude in the stratosphere. MIPAS $CCl_4$ measurements have been compared with independent measurements from other satellite and balloon-borne remote sounders showing a good agreement between the different datasets.

$CCl_4$ trends are calculated as a function of both latitude and altitude. Negative trends of about $-10/-15$ pptv/decade ($-10/-30$ %/decade) are found at all latitudes in the upper-troposphere / lower-stratosphere region, apart from a region in the Southern mid-latitudes between 50 and 10 hPa where the trend is positive with values around $5/10$ pptv/decade ($15/20$ %/decade). At the lowest altitudes sounded by MIPAS, we find trends consistent with those determined on the basis of long-term ground-based measurements ($-10/-13$ pptv/decade). For higher altitudes, the trend shows a pronounced asymmetry between Northern and Southern hemispheres, and the magnitude of the decline rate increases with altitude.

We use a simplified model assuming tracer-tracer linear correlations to determine $CCl_4$ lifetime in the lower stratosphere. The calculation provides a global average lifetime of 47(39 - 61) years considering CFC-11 as the reference tracer. This value is consistent with the most recent literature result of 44(36 - 58) years.

# 1 Introduction

Carbon tetrachloride ($CCl_4$) is a strong ozone-depleting substance with an ozone depletion potential of 0.72 and a strong greenhouse gas with a 100-year global warming potential of 1730 (Harris et al., 2014). Regulated by the Montreal Protocol, the production of $CCl_4$ for dispersive applications was banned for developed countries in 1996, while developing countries were allowed a delayed reduction with the complete elimination by 2010 (Liang et al., 2014). $CCl_4$ can still be legally used as a feedstock, for example in the production of hydro-fluorocarbons. $CCl_4$ natural emissions are not completely understood, which yields some uncertainty on the magnitude of their contributions. Stratospheric Processes and their Role in Climate (SPARC) community (SPARC, 2016) has recently defined an upper limit of the natural emissions (based on the analysis of old air in firn snow) of 3-4 Gg yr$^{-1}$ out of a total emission estimation of 40 (25-55) Gg yr$^{-1}$.

The dominant loss mechanism for atmospheric $CCl_4$ is through photolysis in the stratosphere. The other major sinks are degradation in the oceans and degradation in soil. The estimated partial lifetimes provided in the latest ozone assessment report (Carpenter et al., 2014) with respect to these three sinks are 44 years for the atmospheric sink, 94 years for the oceanic sink, and 195 years for the soil sink. The combination of these three partial loss rates yields a total lifetime estimate of 26 years.

$CCl_4$ atmospheric concentration is routinely monitored by global networks such as Advanced Global Atmospheric Gases Experiment (AGAGE, http://agage.mit.edu/) (Simmonds et al., 1998; Prinn et al., 2000, 2016) and National Oceanic and Atmospheric Administration / Earth System Research Laboratory / Halocarbons & other Atmospheric Trace Species (NOAA / ESRL / HATS, http://www.esrl.noaa.gov/gmd/hats/). The concentration of $CCl_4$ has been decreasing in the atmosphere since the early 1990s, and the latest ozone assessment report (Carpenter et al., 2014) indicates that the global surface mean mole fraction of $CCl_4$ continued to decline from 2008 to 2012. AGAGE and University of California Irvine (UCI) networks report rates of decline of 1.2–1.3% yr$^{-1}$ from 2011 to 2012, whereas the rate of decline reported by the NOAA/HATS network was 1.6% yr$^{-1}$. These relative declines in mole fractions at the Earth's surface are comparable to declines in column abundances of 1.1–1.2% yr$^{-1}$ (Brown et al., 2011; Rinsland et al., 2012).

A significant discrepancy is observed between global emissions estimates of $CCl_4$ derived by reported production and feedstock usage (bottom-up emissions) compared to those derived by atmospheric observations (top-down emissions). This discrepancy has recently stimulated a particular interest in furthering the understanding of atmospheric $CCl_4$. A study performed with a 3-D chemistry-climate model using the observed global trend and the observed inter-hemispheric gradient ($1.5 \pm 0.2$ ppt for 2000–2012) estimated a total lifetime of 35 years (Liang et al., 2014). Recently, a study has reassessed the partial lifetime with respect to the soil sink to be 375 years (Rhew and Happell, 2016), and another study has reassessed the partial lifetime with respect to the ocean sink to be 209 years (Butler et al., 2016). These new estimates of the partial lifetimes with respect to soil and oceanic sinks produce a new total lifetime estimate of 33 years, consistent with the estimate given in Liang et al. (2014). This longer total lifetime reduces the discrepancy between the bottom-up and top-down emissions from 54 Gg yr$^{-1}$ to 15 Gg yr$^{-1}$ (SPARC, 2016). While the new bottom-up emission is still less than the top-down emission, the new estimates reconcile the $CCl_4$ budget discrepancy when considered at the edges of their uncertainties. A recent study estimated that the

average European emissions for 2006–2014 were 2.3 Gg $\text{yr}^{-1}$ (Graziosi et al., 2016), with an average decreasing trend of 7.3% per year.

Since the atmospheric loss of $CCl_4$ is mainly due to photolysis in the stratosphere, satellite measurements that provide vertical profiles are particularly useful in validating the stratospheric loss rates in atmospheric models. A global distribution
of $CCl_4$ extending up to the mid-stratosphere was obtained by the Atmospheric Chemistry Experiment-Fourier Transform Spectrometer (ACE-FTS) (Allen et al., 2009). This study derived an atmospheric lifetime of 34 years through correlation with CFC-11. Another study using ACE-FTS measurements in Brown et al. (2011) estimated the $CCl_4$ atmospheric lifetime to be 35 years. A trend of atmospheric $CCl_4$ from ACE-FTS measurements was reported in Brown et al. (2013), averaged in the 30° S-30° N latitude belt and in the altitude range from 5 to 17 km, where it was found to be decreasing at a rate of 1.2% $\text{yr}^{-1}$.

In this paper, we report the global atmospheric distribution of $CCl_4$ as a function of altitude and latitude obtained from the measurements of the limb emission sounder MIPAS (Michelson Interferometer for Passive Atmospheric Sounding) (Fischer et al., 2008) onboard the ENVISAT satellite. The data product employed here was generated with the processor of the European Space Agency (ESA) Version 7 (ESA, 2016). MIPAS $CCl_4$ vertical profiles are compared with correlative independent measurements. The trend of $CCl_4$ as a function of altitude and latitude is also determined. The MIPAS measurements provide a
denser and more complete geographical coverage than those provided by the ACE-FTS measurements, allowing a more precise knowledge of the $CCl_4$ global distribution and of the trend. The key photolytic loss region (70-20 hPa) is also analyzed.

In Section 2, we introduce MIPAS measurements, the retrieval setup, and the error budget of the $CCl_4$ profiles. In Section 3, we discuss the global $CCl_4$ distribution and the inter–hemispheric differences determined from MIPAS measurements. In Section 4, we show the results of the comparisons between MIPAS and $CCl_4$ correlative measurements from the balloon
version of the MIPAS instrument and the ACE-FTS. In Section 5, we illustrate the method adopted for the estimation of the atmospheric trends and the results of trend analysis, along with some comparisons to previously published results. In Section 6, we evaluate the $CCl_4$ stratospheric lifetime using the tracer-tracer linear correlation method and compare the results with previously published estimates.

## 2 MIPAS measurements

In the first two years of operation (from July 2002 to March 2004) MIPAS acquired, nearly continuously, measurements at Full spectral Resolution (FR), with a spectral sampling of $0.025\ \text{cm}^{-1}$. On 26 March 2004, FR measurements were interrupted due to an anomaly in the movement of the interferometer drive unit. After instrument diagnosis and tests by the hardware experts, atmospheric measurements were resumed in January 2005. After this date, however, MIPAS adopted a reduced spectral resolution of $0.0625\ \text{cm}^{-1}$. Being achievable with a shorter interferometric scan, measurements with this spectral resolution
require a reduced measurement time compared to the FR, thus allowing a finer spatial sampling. For this reason, the measurements acquired from January 2005 onward are referred to as Optimized Resolution (OR) measurements. Compared to the FR, they show both a reduced Noise Equivalent Spectral Radiance (NESR), and finer vertical and horizontal spatial samplings. The nominal FR (OR) scan pattern consists of 17 (27) sweeps with tangent heights in the range from 6-68 (7-72) km with 3 (1.5) km

steps in the Upper Troposphere / Lower Stratosphere (UTLS) region. Full details of the MIPAS measurements acquired in the two mission phases are reported in Raspollini et al. (2013). It is worth mentioning here that in both mission phases MIPAS measurements cover the whole globe with a dense sampling, allowing the study of the evolution of atmospheric composition in great detail. The ESA operational Level 2 algorithm retrieves target parameters at the tangent points of the limb measurements (or at a subset of them). The inversion process minimizes the $\chi^2$–function, using the Gauss-Newton iterative scheme with the Marquardt modification. An adaptive a-posteriori regularization is used in order to smooth the profiles with a strength determined on the basis of the error bars of the unregularized profile (Ceccherini, 2005; Ceccherini et al., 2007; Ridolfi and Sgheri, 2009, 2011). The ESA Level 2 processor version 7 retrieves $CCl_4$ volume mixing ratio (VMR) profiles simultaneously with a set of other target parameters. The retrieval is based on the fit of a set of narrow ($3 \text{ cm}^{-1}$) spectral intervals called microwindows (MWs) containing relevant information on the target parameters. As for all MIPAS ESA retrievals, the MWs for $CCl_4$ retrievals are selected with the MWMAKE algorithm (Dudhia et al., 2002). This algorithm identifies the spectral intervals to be used in the inversion, with the aim of minimizing the total retrieval errors (including both systematic and random components). The MWs used in the ESA Level 2 retrievals from nominal FR and OR measurements are listed in Table 1.

$CCl_4$ VMR is retrieved only up to about 27 km, since above this altitude the $CCl_4$ concentration is too small to generate a sufficient contribution to the measured spectrum for analysis. Moreover OR measurements sample the limb with a vertical step of 1.5 km, significantly finer than the instrument Field Of View ($\approx$3 km). For this reason, to avoid numerical instabilities due to oversampling, in the inversion of OR measurements the retrieval grid includes only one out of every two tangent points. Fig. 1 characterizes a typical $CCl_4$ retrieval from nominal limb scans acquired in the FR (top panel) and OR (bottom panel) measurement phases. The coloured solid lines show the rows of the Averaging Kernels (AKs), each row corresponding to a retrieval grid point (8 grid points for FR and 7 for OR retrievals). Typically the number of degrees of freedom of the retrieval (trace of the AK matrix) is 5–6 for FR and 4–5 for OR measurements. The slightly smaller number of degrees of freedom obtained in the OR retrievals stems from the fact that, to make the retrieval more stable, $CCl_4$ is not retrieved at every tangent point of the OR limb measurements. The dotted red line of Fig. 1 represents the vertical resolution, calculated as the Full Width Half Maximum (FWHM) of the AK rows.

## 2.1 Error budget

To evaluate the $CCl_4$ VMR error due to the mapping of the measurement noise in the retrieval we use the error covariance matrix provided by the retrieval algorithm (Ceccherini and Ridolfi, 2010). The other error components affecting the individual $CCl_4$ VMR profiles are evaluated at Oxford University using the MWMAKE tool. Fig. 2 summarizes the most relevant error components affecting each individual retrieved $CCl_4$ profile, using the MWs of Table 1, for both the FR (top panel) and OR (bottom panel) nominal MIPAS measurement cases.

The key "RND" in the plots refers to the mapping of the measurement noise in the retrieval, as evaluated for typical FR and OR retrievals. Apart from the "NLGAIN" error that will be discussed later, the other error components, in both the FR and OR cases, can be grouped as follows: a) the errors due to the uncertainties in the (previously retrieved) pressure and temperature profiles (PT), and VMR of spectrally interfering gases, for example $O_3$, $H_2O$, $HNO_3$ and $NH_3$; b) the error due to horizontal

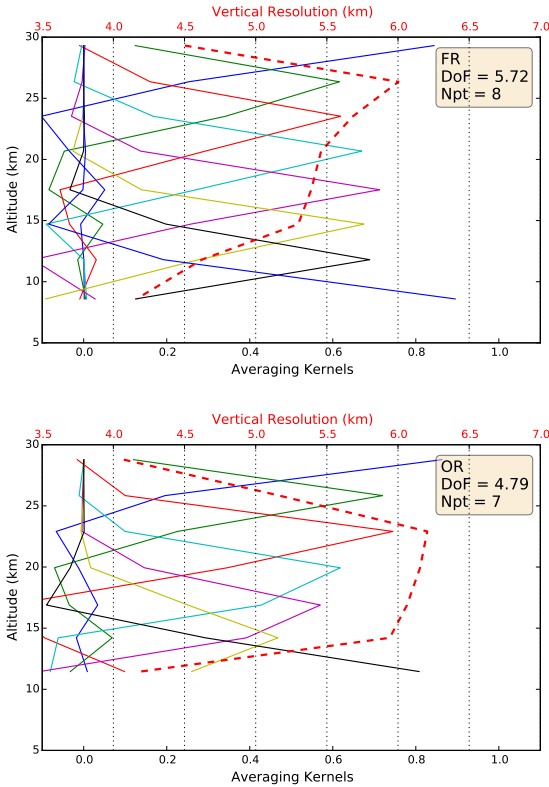

**Figure 1.** Typical Averaging Kernels (AKs, coloured solid lines) and vertical resolution (red dotted lines) of CCl4 VMR retrieved from Full Resolution (FR, top) and Optimized Resolution (OR, bottom) MIPAS measurements. The vertical resolution is calculated as the FWHM of the AK rows. The plot's key shows also the average number of degrees of freedom (DoF) of the retrieval (trace of the AK matrix) and the number of retrieval grid points (Npt).

variability of the atmosphere (GRAD) not included in the model; c) the uncertainties in the spectroscopic (SPECDB) and cross-section (LUT) databases and the error in the $CO_2$ line mixing model (CO2MIX); d) the errors due to less than perfect instrument line-shape characterization, namely its spectral shift (SHIFT) and width (SPREAD). For the details on how the different error components were calculated by MWMAKE, see Dudhia et al. (2002) and the Oxford University MIPAS website (Oxford University, 2016).

The main errors of type a) are due to interfering gases whose VMRs are retrieved before $CCl_4$ with some random error. Therefore, like the RND error component, they change randomly from profile to profile. Thus, in the calculated (monthly) averages they scale down with the inverse square root of the number of averaged profiles. The errors of type b), as shown

| MWs used in CCl$_4$ retrievals from FR measurements | |
|---|---|
| Start wavenumber cm$^{-1}$ | End wavenumber cm$^{-1}$ |
| 796.3750 | 799.3750 |
| 800.2750 | 803.2750 |
| 792.7000 | 795.7000 |
| 771.8000 | 773.7750 |

| MWs used in CCl$_4$ retrievals from OR measurements | |
|---|---|
| Start wavenumber cm$^{-1}$ | End wavenumber cm$^{-1}$ |
| 792.8125 | 795.8125 |

**Table 1.** Microwindows (MWs) used for CCl$_4$ retrieval from nominal FR and OR MIPAS measurements.

in Castelli et al. (2016), cause systematic (and opposite in sign) differences between profiles retrieved from measurements acquired in the ascending and the descending parts of the satellite orbits. These errors largely cancel out when calculating averages that evenly include profiles retrieved from measurements belonging to the ascending and the descending parts of the orbits. Errors of type c) are constant and may cause profile biases but have no effect on calculated trends. Regarding the errors

due to the imperfect instrument line-shape modeling (type d), since the gain of MIPAS bolometric detectors remained constant throughout the whole mission, there is no hint of a possible degradation of instrument optics and thus of a possible change in the instrument line-shape. This type of error, therefore, has no impact on the trend calculation.

Imperfect instrument radiometric calibration also causes an error. This error is plotted in Fig. 2 with the label "NLGAIN". Being of the order of $0.4\%$ in the upper part of the retrieval range, it is rather small in individual CCl$_4$ profiles. Although

small, this error is important when calculating atmospheric trends as it includes the uncertainty in the correction applied to the radiances to account for the non-linearities of MIPAS photometric detectors (Kleinert et al., 2007). In MIPAS Level 1b radiances up to version 5, the applied non-linearity correction is constant throughout the whole MIPAS mission. However, non-linearities change over the course of the mission due to progressive ageing of the detectors. A constant correction implies, therefore, a drift of the radiometric calibration error during the mission, with a direct impact in the calculated trends. MIPAS

Level 1b radiances version 7 overcome this problem as they use a time-dependent non-linearity correction scheme. The residual drift of the calibration error after this time-dependent correction is still being characterized; however, preliminary results (Birk priv. com. 2016) show that it is smaller than 1% across the entire mission. MIPAS Level 1b radiances version 5 were used in the past to extract information on trends of different gases, either ignoring this effect (see, e.g., CFC-11/CFC-12 in Kellmann et al. (2012), or HCFC-22 in Chirkov et al. (2016)) or correcting the drift via intercomparison with other instruments assumed

to be drift-free (Eckert et al., 2014). Recently it has been shown (Eckert et al., 2016) that ignoring this effect introduces a significant error on the trend estimation. The MIPAS Level 1b calibrated radiances version 7 employed here are considered to be a significant improvement from the point of view of the correction of this drift.

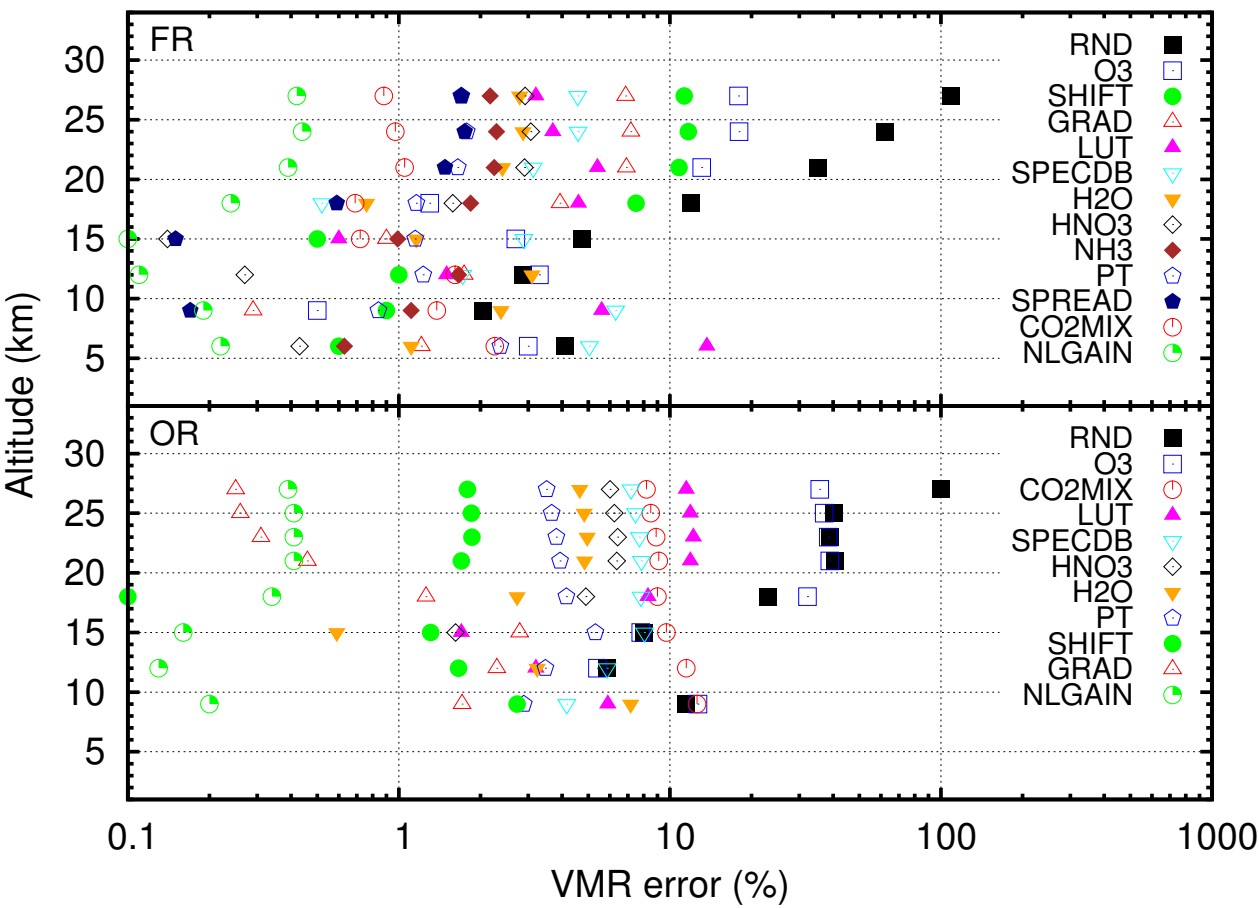

**Figure 2.** Main error components of the individual retrieved $CCl_4$ VMR profiles from FR (top) and OR (bottom) nominal MIPAS measurements.

The generally good quality of fits obtained in $CCl_4$ retrievals is illustrated in Fig. 3. The figure refers to the MWs used in the FR retrievals. We do not show the residuals in the single MW used for OR retrievals as it mostly overlaps the third MW of FR retrievals. The upper plot of Fig. 3 shows the average of 1141 observed (black dots) and simulated (red line) limb radiances in the MWs used for $CCl_4$ retrievals. The averages include spectra with tangent heights in the range from 6 to 17 km. The lower plot shows the average residuals of the fit (observation minus simulation, blue line) as well as the average noise level of the individual MIPAS measurements (dashed lines). The grey areas indicate spectral channels that, as recommended by the MWMAKE algorithm, are excluded from the fit to minimize the total retrieval error. Note that the average residuals shown in Fig. 3 have an associated random error given by the noise of the individual measured spectra divided by the square root of the number of averaged spectra, i.e. $\approx 1 nW/(cm^2 sr\, cm^{-1})$. This implies that while the magnitude of the average residuals is

incompatible with their noise error, the additional systematic uncertainties are still smaller than the noise error of the individual measured spectra, in agreement with the predictions reported in Fig. 2.

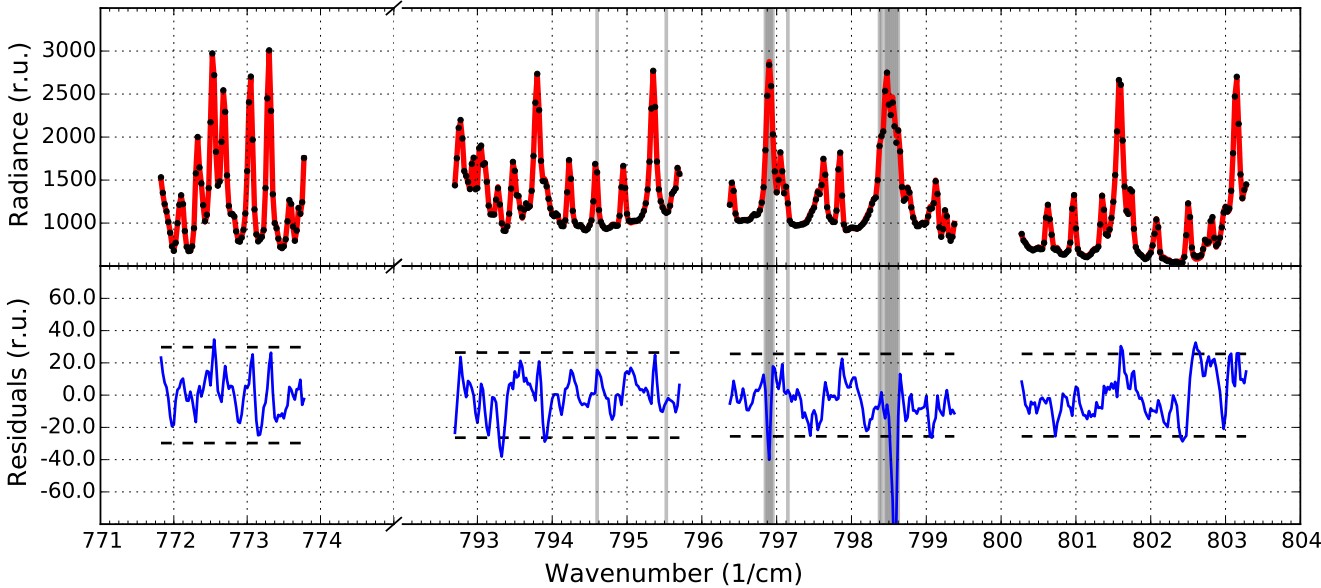

**Figure 3.** The upper plot shows an average of 1141 observed (black dots) and simulated (red line) limb radiances in the MWs used for $CCl_4$ FR retrievals. The averages include spectra with tangent heights from 6 to 17 km. The lower plot shows the average residuals of the fit (blue line, observation minus simulation) as well as the average noise level of the individual measurements (dashed lines). The grey areas indicate spectral channels excluded from the fit. The radiance units (r.u.) in the vertical axes of the plots are $nW/(cm^2 sr\, cm^{-1})$.

## 3 $CCl_4$ global distribution

Figure 4 shows the global monthly distribution of MIPAS $CCl_4$ VMR for a representative month from each of the four seasons, spanning the time period from August 2010 through May 2011. Here, retrieved profiles were first interpolated to fixed pressure levels (see Sect. 5.1), and then binned in 5° latitude intervals. In all the considered months, the zonal averages show the typical shape of long-lived species of anthropogenic origin, which are emitted at the surface and destroyed primarily in the stratosphere. Larger values are found in the troposphere, and then the VMR monotonically decreases with increasing altitude in the stratosphere. In the lower stratosphere, concentrations between 30° S and 30° N are significantly larger compared to those at higher latitudes. This pattern can be attributed to the Brewer-Dobson circulation that is responsible for the uplift of the surface air in the tropical regions.

The maps in Fig. 5 show the time evolution of $CCl_4$ at all latitudes from July 2002 to April 2012. The three maps refer to different pressure levels: 50 hPa (upper map), 90 hPa (middle map) and 130 hPa (lower map). The $CCl_4$ time evolution maps show a seasonal variability. The intrusion of $CCl_4$-poor mesospheric air in the stratosphere during winter, due to the

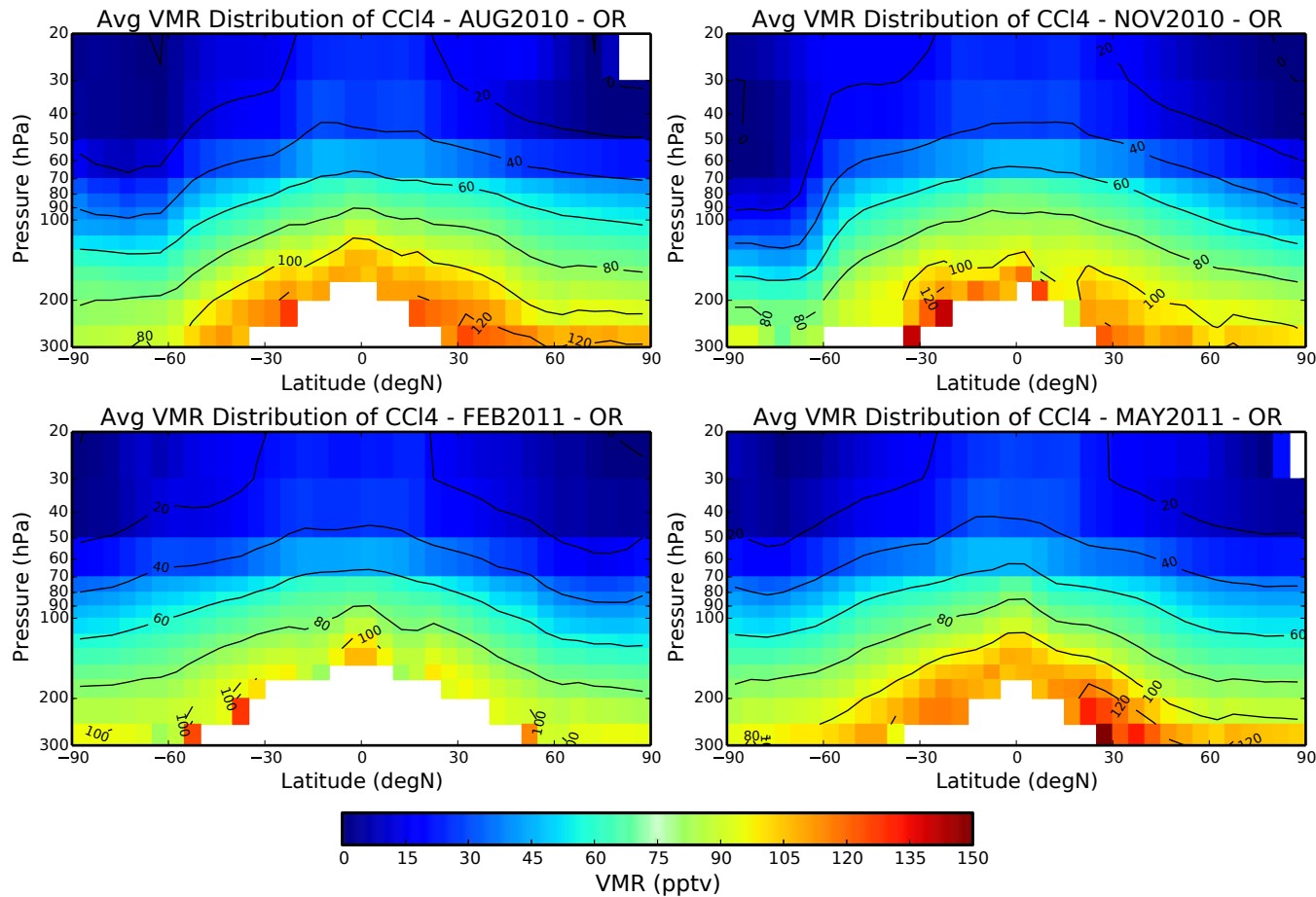

**Figure 4.** Zonal monthly averages of MIPAS $CCl_4$ profiles. The maps refer to four separate months in different seasons: August 2010 (top left), November 2010 (top right), February 2011 (bottom left) and May 2011 (bottom right).

air subsidence induced by the polar vortex, is clearly visible in both polar winters, its effects continuing into early spring and extending into the troposphere. Minimum $CCl_4$ values are observed in November at the South Pole and in March at the North Pole (November is considered the beginning of spring at the South Pole, whereas spring begins in March at the North Pole). This was previously observed for other long lived anthropogenic species (Kellmann et al., 2012). The effect is larger in the

5   Antarctic due to the stronger, more stable polar vortex.

Modulated by this seasonal variability, at all altitudes a constant trend and an inter-hemispherical difference can also be observed and are further analysed in the subsequent figures. We also note that for pressures larger than 100 hPa, the $CCl_4$ measured in the OR phase has a positive bias with respect to that measured in the FR phase. This bias, discussed also in Sect. 4.1, may be due to the different MWs used for the retrieval in the two mission phases, or to the different limb sampling

10   patterns adopted.

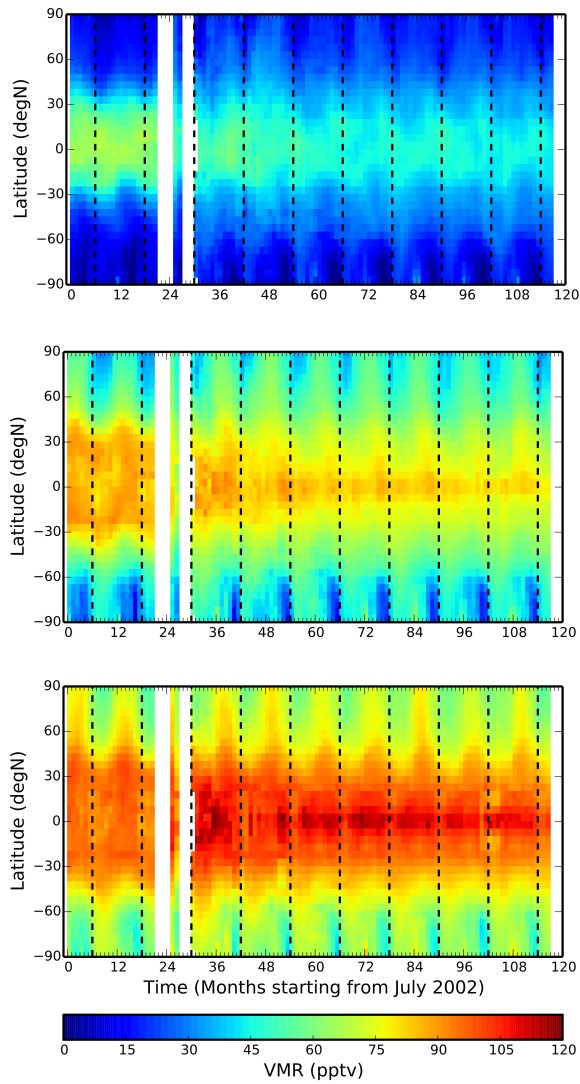

**Figure 5.** Time evolution of $CCl_4$ at all latitudes, from July 2002 to April 2012. The three maps refer to different pressure levels: 50 hPa (top), 90 hPa (center) and 130 hPa (bottom). The vertical dashed lines represent the year boundaries.

The Inter Hemispheric Gradient (IHG) at the surface is largely used as a qualitative indicator of continuous emissions (Lovelock et al., 1973; Liang et al., 2014). Anthropogenic emissions are larger in the Northern Hemisphere (NH) (SPARC, 2016) and the transport of these emissions from the NH to the Southern Hemisphere (SH) takes about one year, i.e. a time interval much shorter than the $CCl_4$ lifetime (see Sect. 6). Hence, a significant IHG in the $CCl_4$ distribution represents evidence of ongoing emissions.

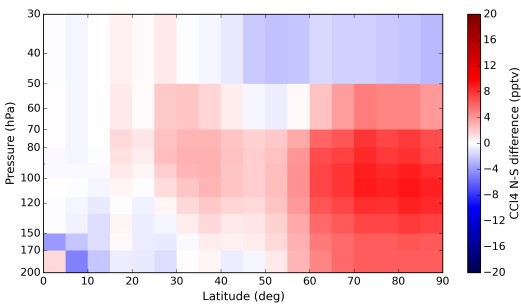

**Figure 6.** Average North-South $CCl_4$ VMR differences versus latitude and pressure. The average period includes MIPAS measurements from April $1^{st}$, 2005 to March $31^{st}$, 2012.

Although MIPAS measurements are not suitable to evaluate the IHG at the surface, they provide information about the distribution of inter-hemispheric differences in the UTLS region as a function of both latitude and pressure. To analyze these differences we interpolated to a fixed pressure grid MIPAS $CCl_4$ profiles acquired from April 2005 to March 2012. We then binned the profiles in $5°$ latitude intervals and calculated, for each latitude bin, the average $CCl_4$ VMR profile in the considered
5   time period. Finally, for each latitude bin in the NH we identified the corresponding bin in the SH and computed the difference between the average profiles. The map of Fig. 6 shows the obtained average differences as a function of both latitude bin and pressure level. At high latitudes, the asymmetry likely stems from the fact that the polar vortex in the Antarctic is systematically stronger, more stable, and of longer duration than the Arctic polar vortex. At mid-latitudes, NH and SH seasons are more symmetrical and the $CCl_4$ mean differences between the two hemispheres are probably caused by the larger $CCl_4$ emissions
10   in the NH (SPARC, 2016; Liang et al., 2014).

As a final test we computed the weighted average of the NH-SH differences over latitude at fixed pressure levels. The weights used in the average are the solid angle fractions viewed by the individual latitude bands. The NH-SH mean differences in the UTLS span from 1.2 ppt at 130 hPa to 2.2 ppt at 100 hPa. At the lowermost pressure levels these differences are fully consistent with the IHG value of $1.5 \pm 0.2$ ppt (for 2000-2012) reported by Liang et al. (2014).

15   **4   Comparison to other $CCl_4$ measurements**

The most accurate atmospheric $CCl_4$ measurements are collected at ground level, but such measurements are not suitable for direct comparison with profiles retrieved from MIPAS measurements in the 5-27 km height range. In the next two sub-sections we compare MIPAS $CCl_4$ profiles with co-located profiles obtained from the stratospheric balloon version of MIPAS (MIPAS-B, Friedl-Vallon et al. (2004)) and from the ACE-FTS onboard the SciSat-1 satellite (Bernath et al., 2005).

| Location | Date | Distance (km) | Time difference (min) |
|---|---|---|---|
| Kiruna (68 N) | 20 Mar 2003 | 16/546 | 14/15 |
| | 03 Jul 2003 | Trajectories only | |
| | 11 Mar 2009 | 187/248 | 5/6 |
| | 24 Jan 2010 | 109/302 | 5/6 |
| | 31 Mar 2011 | Trajectories only | |
| Aire-sur-l'Adour (44 N) | 24 Sep 2002 | 21/588/410/146 | 12/13/15/16 |
| Teresina (5 S) | 14 Jun 2005 | 109/497/184/338 | 228/229/268/269 |
| | 06 Jun 2008 | 224/284/600/194 | 157/158/169/170 |

**Table 2.** Overview of MIPAS balloon flights used for intercomparison with MIPAS/ENVISAT

## 4.1 Comparison with MIPAS balloon

The balloon-borne limb emission sounder MIPAS-B can be regarded as a precursor of the MIPAS satellite instrument (Friedl-Vallon et al. (2004) and references therein). Indeed, a number of specifications like spectral resolution ($0.0345$ cm$^{-1}$) and spectral coverage ($750$–$2500$ cm$^{-1}$) are similar. However, for other parameters the MIPAS-B performance is superior, in particular for the NESR and for the line of sight stabilization, which is based on an inertial navigation system supplemented with an additional star reference system and leads to a knowledge of the tangent altitude on the order of 90 m ($3\sigma$). The MIPAS-B NESR is further improved by averaging multiple spectra recorded at the same elevation angle. MIPAS-B limb scans are typically acquired on a $1.5$ km vertical tangent height grid.

Retrieval of all species is performed on a 1 km grid with a least squares fitting algorithm using analytical derivative spectra calculated by the Karlsruhe Optimized and Precise Radiative transfer Algorithm (Höpfner et al., 2002; Stiller et al., 2002). To avoid retrieval instabilities due to oversampling of vertical grid points, a regularization approach is adopted, constraining with respect to a first derivative a priori profile according to the method described by Tikhonov and Phillips. The spectral window used for the MIPAS-B target parameter retrieval of $CCl_4$ covers the $786.0$–$806.0$ cm$^{-1}$ interval. Spectroscopic parameters for the calculation of the infrared emission spectra are a combination of the HITRAN 2008 (Rothman et al., 2009) database and the MIPAS dedicated database (Raspollini et al., 2013; Perrin et al., 2016). The $CCl_4$ cross sections are taken from HITRAN as in MIPAS/ESA retrievals version 7. The MIPAS-B error budget includes random noise as well as covariance effects of the fitted parameters, temperature errors, pointing inaccuracies, errors of non-simultaneously fitted interfering species, and spectroscopic data errors ($1\sigma$). For $CCl_4$ the precision error is estimated to be between 5-10%, while the total error is 11-15%. Further details on the MIPAS-B data analysis and error estimation are provided in Wetzel et al. (2012) and references therein. Table 2 lists all the MIPAS-B flights used for intercomparison with MIPAS on ENVISAT.

Further to the direct matches where the balloon and the satellite instruments observe simultaneously (within pre-defined margins) the same air-masses, we also considered trajectory matches. In this case both forward and backward trajectories were

calculated (Naujokat and Grunow, 2003) by the Free University of Berlin from the balloon measurement geolocation to search for air-masses sounded by the satellite instrument. Temperature and VMR values from the satellite profiles were interpolated to the trajectory match altitude such that these values can be directly compared to the MIPAS-B data at the trajectory start point altitude. To identify both direct and trajectory matches, a coincidence criterion of 1 hour and 500 km was adopted.

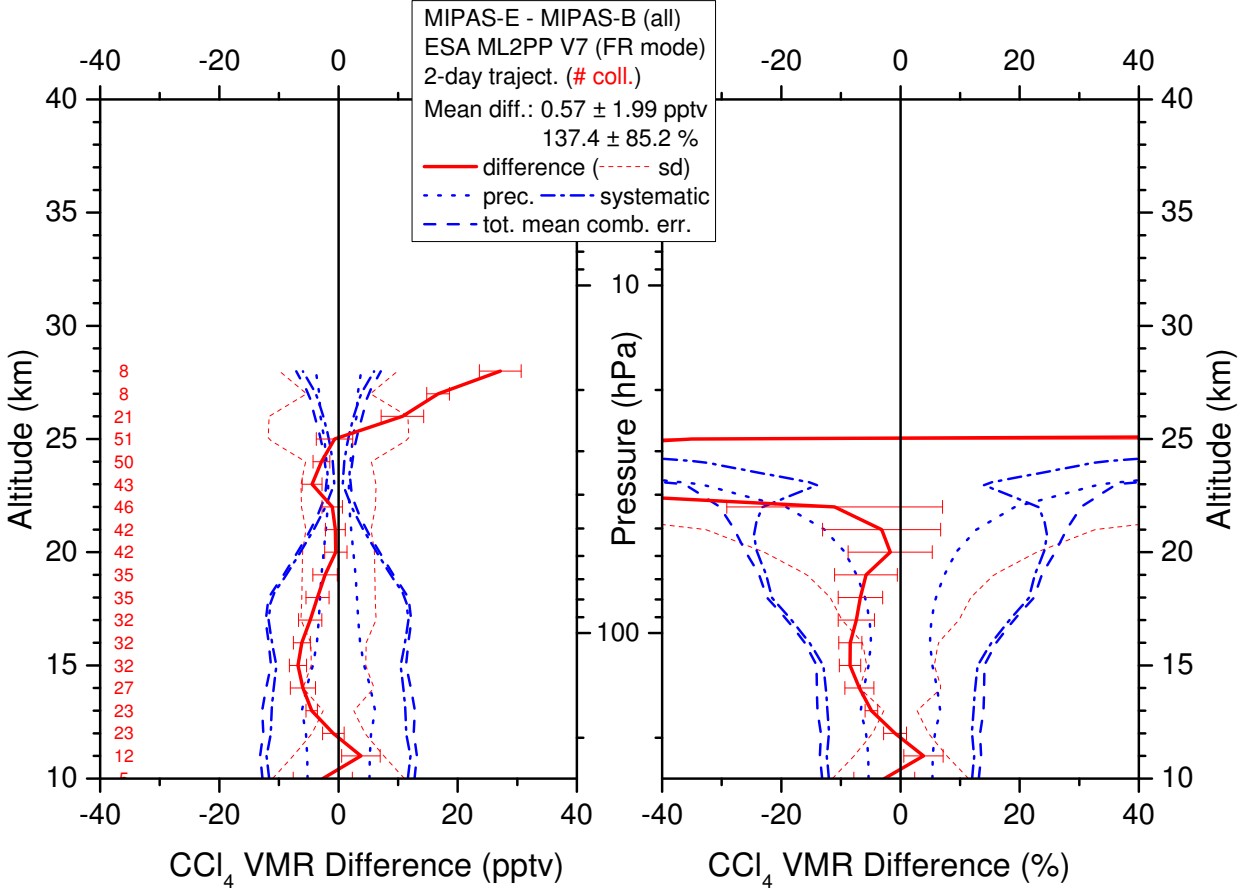

**Figure 7.** Intercomparison between MIPAS-B and MIPAS/ENVISAT (MIPAS-E) CCl4 VMR. Results for the FR part of the MIPAS mission. The plots show mean absolute and relative VMR differences of trajectory match collocations (red numbers) between both MIPAS sensors (red solid line) including standard deviation of the difference (red dotted lines) and standard error of the mean (plotted as error bars). Precision (blue dotted lines), systematic (blue dash-dotted lines) and total (blue dashed lines) mean combined errors calculated according to the error summation ($\sqrt{\sigma^2_{MIPAS-E} + \sigma^2_{MIPAS-B}}$) are also displayed. For further details on the error calculation, see Wetzel et al. (2013).

5    Figures 7 and 8 show the average differences between CCl4 VMR retrieved from MIPAS/ENVISAT and MIPAS-B both in absolute and relative units. The two figures refer to matching measurements in the FR and the OR phases of the MI-PAS/ENVISAT mission, respectively. Combined random, systematic and total errors are also shown in the plots. The numbers

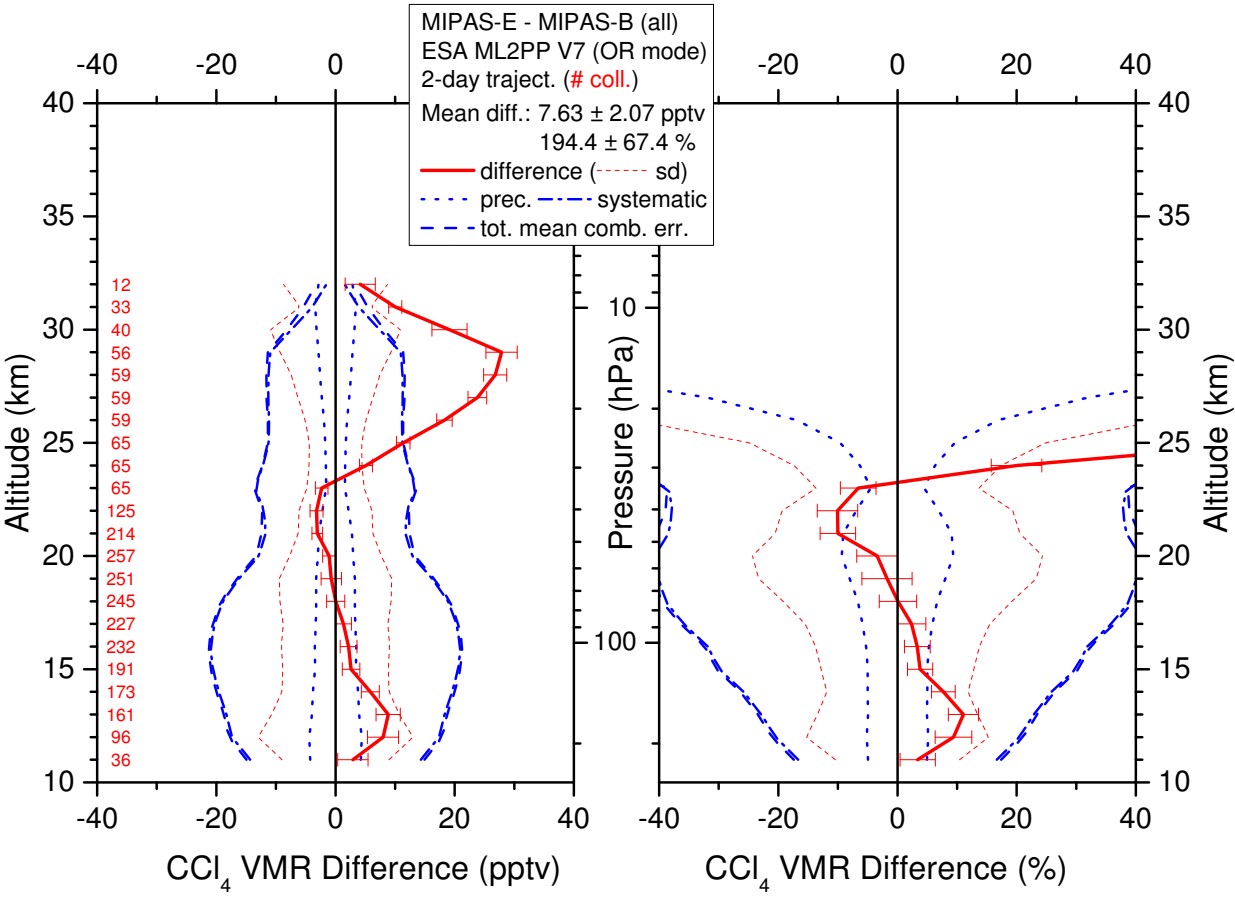

**Figure 8.** Same as Figure 7 but for the OR part of the MIPAS mission.

reported on the left side of the plots indicate the number of matching profiles contributing to the statistics. The results of the intercomparison can be summarized as follows. In the case of FR measurements: for pressures between 80 and 190 hPa MIPAS/ENVISAT shows a statistically significant negative bias of about $-10\%$ with respect to MIPAS-B, this bias is however within the combined total error bounds. A statistically significant positive bias is also evident for pressures smaller than

5  25 hPa. It increases with altitude and quickly becomes incompatible with the total combined error. This bias can be at least partly explained by the selection of different microwindows used during the retrieval process of both MIPAS sensors. This bias, however, is not a major concern because it is localized at the upper end of the retrieval range. In this region the predicted uncertainty is so large that the linear approximation of the error propagation theory may easily fail to explain the discrepancies between the measurements of the two instruments. In case of OR measurements: for pressures between 150 and 190 hPa

10  MIPAS/ENVISAT shows a statistically significant positive bias of about $+10\%$ with respect to MIPAS-B; this bias is however within the combined total error bounds. A statistically significant positive bias is also evident for pressures smaller than 25 hPa.

It increases with altitude and, for pressures smaller than 20 hPa is no longer compatible with the total combined error. As in the FR case, this large bias occurs at the upper end of the MIPAS/ENVISAT retrieval range where the predicted combined error is very large. Furthermore, comparison with ACE (see next Section) indicates a negative bias of MIPAS with respect to ACE-FTS, in the same altitude region, hence MIPAS/ENVISAT is in the middle between MIPAS balloon and ACE-FTS.

## 4.2   Comparison with ACE-FTS V3.5

ACE-FTS is a Canadian solar occultation limb sounder operating since 2004 from SciSat in a low ($\approx 650$ km) circular orbit. The measured spectra cover the region from 750 to 4400 cm$^{-1}$ with a spectral resolution of 0.02 cm$^{-1}$ (Bernath et al., 2005). Several target atmospheric parameters are routinely retrieved from ACE-FTS measurements. Among them, temperature, pressure, and the VMR profiles of over 30 atmospheric trace gases and over 20 subsidiary isotopologues. Profiles are retrieved in the range from $\sim$ 5 to 150 km, with a vertical field of view of $\sim$ 3-4 km and a vertical sampling of 2-6 km. The ACE-FTS retrieval algorithm is described in Boone et al. (2005), and the updates for the most recent version of the retrieval, version 3.5, are detailed in Boone et al. (2013). The retrieval algorithm uses a non-linear least-squares global-fitting technique that fits the ACE-FTS observed spectra in given microwindows with forward modelled spectra based on line strengths and line widths from the HITRAN 2004 database (Rothman et al., 2005) (with updates as described by Boone et al. (2013)). Pressure and temperature profiles used in the forward model are the ACE-FTS derived profiles, calculated by fitting $CO_2$ lines. The spectral window used for $CCl_4$ retrievals extends from 787.5 to 805.5 cm$^{-1}$.

Several hundred ACE-FTS measurements are coincident with MIPAS soundings of the OR part of the mission. These measurements are located both in the Northern and Southern hemispheres, mainly at latitudes larger than 45°. For comparison with MIPAS, all ACE-FTS $CCl_4$ data used were screened using the v3.5 quality flags. As recommended by Sheese et al. (2015), any profile data point with flag value of 2 or greater was removed and any profile containing a flag value between 4 and 7, inclusive, was discarded. For intercomparison with MIPAS measurements we adopted a matching criterion of 3 hours and 300 km. We also tested different matching criteria, such as 2 hours and 300 km, 3 hours and 200 km, but found no significant changes in the intercomparison. First we interpolated the matching MIPAS and ACE-FTS $CCl_4$ profiles to a fixed set of pressure levels. Then we grouped the profile differences in latitudinal intervals. The results of the comparison are summarized in Fig. 9. Each of the four plots of the figure refers to one of the considered latitude intervals: 50–70° and 70–90° in both the Southern and the Northern hemispheres. Each plot shows the average $CCl_4$ difference profile between co-located MIPAS and ACE-FTS measurements (red) with standard deviation of the mean (red error bars, calculated as the standard deviation of the differences divided by the square root of the sample size). The standard deviation of the differences (orange), the total random error (green), the total systematic error of the difference (blue) are also shown. The number of co-located pairs contributing at each pressure level is reported on the right side of each plot. The average difference (red line) quantifies the systematic bias between ACE-FTS and MIPAS, the error bars indicate its statistical significance. The standard deviation (orange) is an *ex-post* estimate of the combined random error of the individual profile differences and, therefore, should be similar to its *ex-ante* estimate represented in the plots by the green line. We calculated the ex-ante random error of the individual profile differences as the quadrature summation of the ACE-FTS and MIPAS random errors. The ACE-FTS random error is estimated

via the noise error covariance matrix of the retrieval included in the Level 2 products. The MIPAS random error is estimated as the quadrature summation of the measurement noise error evaluated by the covariance matrix of the retrieval (Ceccherini and Ridolfi, 2010) and the other error components that are expected to change randomly in our sample, i.e. the errors that we classified of types a) and b) in Sect. 2.1. The systematic error of the profile differences is obtained as the quadrature summation

5    of the ACE-FTS and the MIPAS errors that are constant within the sample and are not expected to bias in the same direction the measurements of the two instruments. On the basis of the error figures suggested by Allen et al. (2009), for ACE-FTS we assumed a 20% systematic error constant at all pressure levels. For MIPAS we calculated the quadrature summation of systematic errors that in Sect. 2.1 we classified as of type c) and d). For the calculation of the combined systematic error we explicitly excluded the uncertainty in the $CCl_4$ cross-section data (Rothman et al., 2005) that are used, approximately in the

10   same spectral region, both in MIPAS and ACE-FTS retrievals.

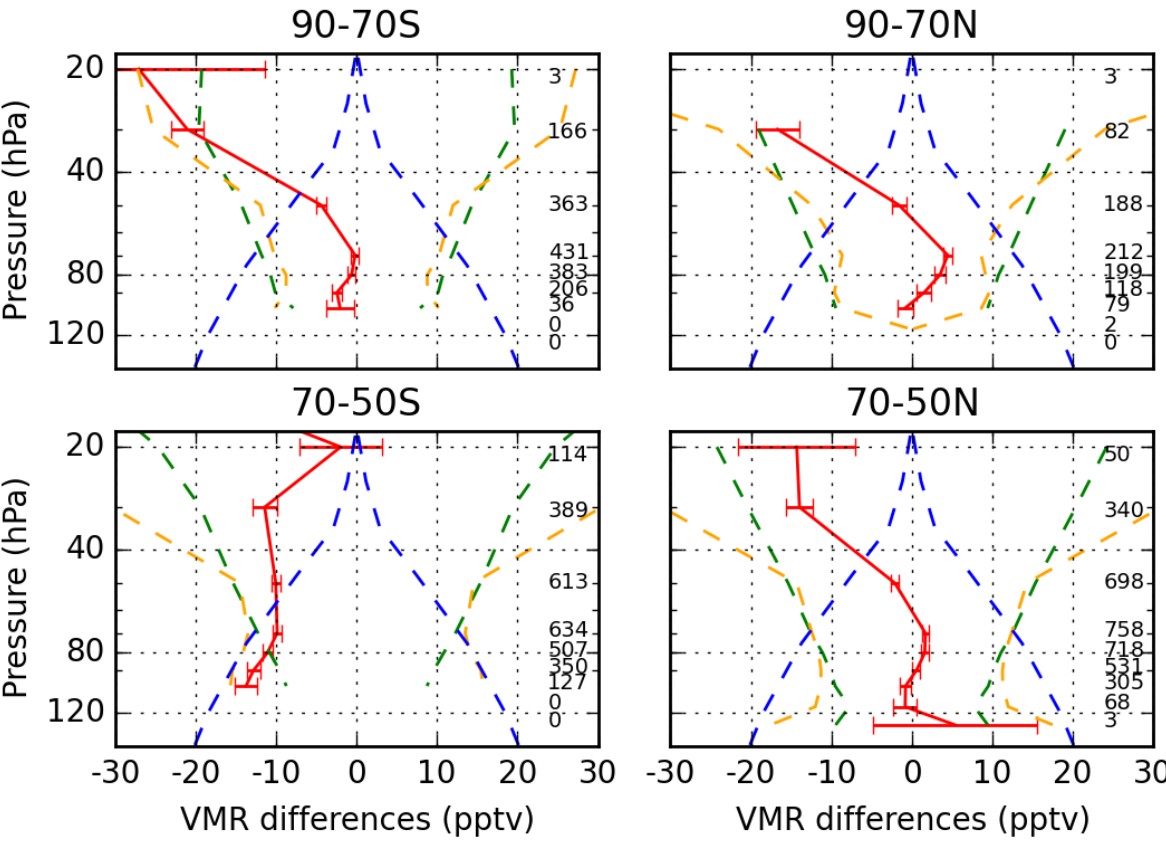

**Figure 9.** Mean $CCl_4$ profile difference between co-located MIPAS and ACE-FTS measurements (red) with standard deviation of the mean (red error bars). The standard deviation of the differences (orange), the estimated total random (green) and total systematic (blue) errors of the difference are also shown. The number of co-located pairs for each pressure level is reported on the right side of each graph. Each plot refers to a latitude interval as indicated in the title.

Apart from the latitude interval from 50 to 70° S, the systematic differences between MIPAS and ACE-FTS are within 5 pptv ($\sim 10$ %, mostly not significant from the statistical point of view) in the pressure range from 50 to 100–110 hPa. The amplitude of systematic differences increases up to 15–20 pptv and becomes statistically significant at 30 hPa, while it is again quite small at 20 hPa. In the latitude interval from 50 to 70° S we observe a statistically significant $\approx 10$ pptv low bias of MIPAS with respect to ACE-FTS, almost uniform over the entire retrieval height range. At all latitudes, the observed biases are compatible with the estimated combined systematic error only for pressures greater than 40 hPa. At 30 hPa the bias is statistically significant and incompatible with error bars. The reason for this inconsistency is still unclear; however, preliminary investigations show that the inconsistency will be reduced when using the future release version 4.0 of ACE-FTS products.

The ex-ante estimate of the combined random error (green line in Fig. 9) agrees pretty well with the ex-post estimated standard deviation of the profile differences (orange line) in the range between 40 and 80–100 hPa. At the limits of the retrieval range the observed variability of the differences generally exceeds the ex-ante estimate of the random error. This may be due both to the fact that our ex-ante random error estimate does not take into account the imperfect matching of the compared profiles, and to the fact that, at these specific altitudes, the sensitivity of the measurements to the $CCl_4$ VMR is so low that the linear approximation of the error propagation theory could provide only rough error estimates.

As a final remark we note that at 30 hPa MIPAS-B (Fig. 8) and ACE-FTS (Fig. 9) intercomparisons provide contrasting indications on the MIPAS bias in the OR part of the mission. While MIPAS-B suggests a positive MIPAS bias of about 10 pptv, ACE-FTS points to a negative bias of $10 - 20$ pptv.

## 5 Trends

### 5.1 Trend calculation method

The measurements used for the analysis presented in this study cover the entire MIPAS mission, from July 2002 to April 2012. The $CCl_4$ VMR profiles considered are those derived by the ESA Level 2 processor version 7 analysing MIPAS limb scanning measurements with tangent heights in the 6-70 km range, obtained from nominal (NOM), middle atmosphere (MA) and Upper Troposphere Lower Stratosphere (UTLS1) observational modes (Raspollini et al., 2013).

First we linearly interpolate in log-pressure all the considered $CCl_4$ VMR profiles to the 28 SPARC data initiative (Hegglin and Tegtmeier, 2011) pressure levels (300, 250, 200, 170, 150, 130, 115, 100, 90, 80, 70, 50, 30, 20, 15, 10, 7, 5, 3, 2, 1.5, 1.0, 0.7, 0.5, 0.3, 0.2, 0.15, 0.1 hPa). We then group the interpolated profiles in 5° latitude bins and calculate monthly averages. Finally, using the least-squares method, for each latitude bin and pressure level we fit the following function $VMR(t)$ to the time series of the monthly averages:

$$
\begin{aligned}
VMR(t) = a_{FR}\,\mathbf{1}_{FR}(t) + a_{OR}\,\mathbf{1}_{OR}(t) + bt + f_1\,qbo30(t) + f_2\,qbo50(t) + \\
+ g\,SRF(t) + \sum_i \left[ c_i \sin\left(\frac{2\pi t}{T_i}\right) + d_i \cos\left(\frac{2\pi t}{T_i}\right) \right].
\end{aligned}
\tag{1}
$$

In this expression $t$ is the time expressed in months since the beginning of the mission (July 2002) and $a_{FR}$, $a_{OR}$, $b$, $f_1$, $f_2$, $g$ and $c_i, d_i$, $i = 1, ..., 8$ are the 22 fitting parameters. The function $\mathbf{1}_P(t)$ is the indicator function of the time interval P, such that $\mathbf{1}_P(t) = 1$ if $t \in P$ and $\mathbf{1}_P(t) = 0$ otherwise. The functions $\text{qbo}30(t)$ and $\text{qbo}50(t)$ are the quasi-biennial oscillation (QBO) quantifiers and $\text{SRF}(t)$ is the solar radio flux index. The two QBO terms (available at http://www.geo.fu-berlin.de/met/ag/strat/produkte/qbo/index.html) represent the Singapore winds at 30 and 50 hPa (Kyrölä et al., 2010). The $\text{SRF}$ index is calculated using measurements of the solar flux at 10.7 cm (available at http://lasp.colorado.edu/lisird/tss/noaa_radio_flux.html) and is considered a good proxy for the solar activity. We re-normalized both the QBO and the SRF proxies to the interval $[-1, +1]$ within the time frame covered by MIPAS mission. The terms in the sum are 8 sine and 8 cosine functions. They represent periodic oscillations with period $T_i$. In $T_i$ we include annual (12 months), semi-annual (6 months) and other characteristic atmospheric periodicities of 3, 4, 8, 9, 18 and 24 months (Haenel et al., 2015). We decided to fit two different constant parameters for the two parts of the mission: $a_{FR}$ for the FR and $a_{OR}$ for the OR part. The aim of this choice is to account for possible relative biases between the two phases of the mission. These may be caused, for example, by the different spectral resolutions adopted, by the different MWs used for the retrieval and by the different vertical and horizontal samplings of the instrument in the two mission phases. We calculate the uncertainty on the fitted parameters assuming each monthly average is affected by an error given by the standard deviation of the mean. Furthermore we multiply the uncertainty obtained from the error propagation analysis by the square root of the normalized least squares (the so-called "reduced $\chi^2$"). This latter operation is intended to account also for the quality of the fit in the evaluation of trend errors.

## 5.2  Results

Figure 10 shows some examples of $CCl_4$ trend analysis. Each panel refers to a specific latitude band and pressure level. The top plot of each panel shows the time series of the monthly averages with error bars given by the standard deviation of the mean (blue symbols). The red curve represents the best fitting function $\text{VMR}(t)$, while the green line represents the constant and the linear (trend) terms of $\text{VMR}(t)$. In the lower plot of each panel we show the residuals of the fit (the monthly averages minus the values calculated on the fitting curve). In each panel we also report the value obtained for the trend, its uncertainty and the difference between the two constant terms $a_{FR} - a_{OR}$.

The quality of the fit is generally better in the OR period. Indeed, in this mission phase the instrument provides measurements with more uniform and finer geographical coverage. We also carried out a spectral analysis of the fitting residuals, which revealed that all the periodicities embedded in the considered time series of monthly means are properly accounted for by the fitting function (1).

Figure 11 summarizes the results obtained for $CCl_4$ trends. Panel a) shows the absolute trends. Negative trends are observed at all latitudes in the UTLS region. The magnitude of the negative trend decreases with increasing altitude. The trend shows slightly positive values (about 5-10 pptv/decade) in a limited region, particularly in the Southern mid-latitudes between 50 and 10 hPa. This feature is probably related to the asymmetry in the general circulation of the atmosphere. The air at higher altitudes can be considered *older* than the tropospheric air that has been lifted up by strong convection mechanisms in the tropical regions (Stiller et al., 2012). The tropospheric air just injected into the stratosphere is richer in $CCl_4$. We attribute

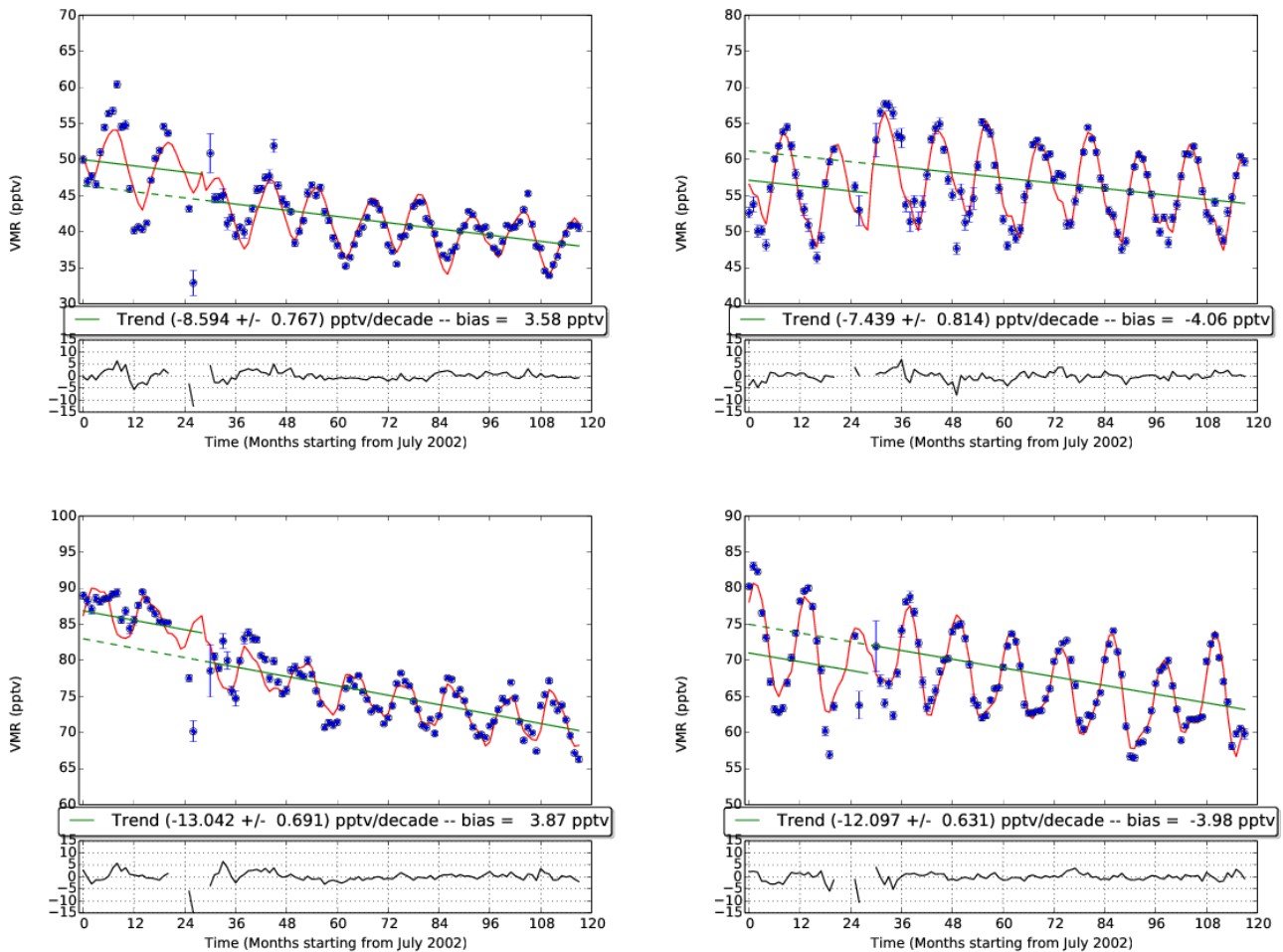

**Figure 10.** CCl$_4$ trend analysis for 20° S/25° S at 50 hPa (top left), 55° S/60° S at 100 hPa (top right), 25° N/20° N at 90 hPa (bottom left) and 50° N/45° N at 100 hPa (bottom right). The blue dots are the MIPAS monthly averages and the error bars are the standard deviation of the means. The red curve is the best fitting function VMR($t$) and green line is the linear term (trend). The lower part of each plot shows the residuals between the MIPAS monthly averages and the best fitting function VMR($t$). The CCl$_4$ trend, its uncertainty and the *bias* between FR and OR are also indicated in each panel.

positive stratospheric trend values in certain latitude regions to the less effective mixing mechanisms in the stratosphere as compared to the troposphere at these latitudes. Similar features have also been observed by other authors in CFC-11 and CFC-12 trends (Kellmann et al., 2012). Recently some studies (Harrison et al., 2016; Mahieu et al., 2014; Ploeger et al., 2015) have shown that the trends in stratospheric trace gases are affected by variability in the stratospheric circulation. This has been shown for a number of halogen source gases and the complementary degradation products (i.e. HCl and HF). This variability can partially explain why the stratospheric trend does not simply follow the tropospheric trend with a time lag.

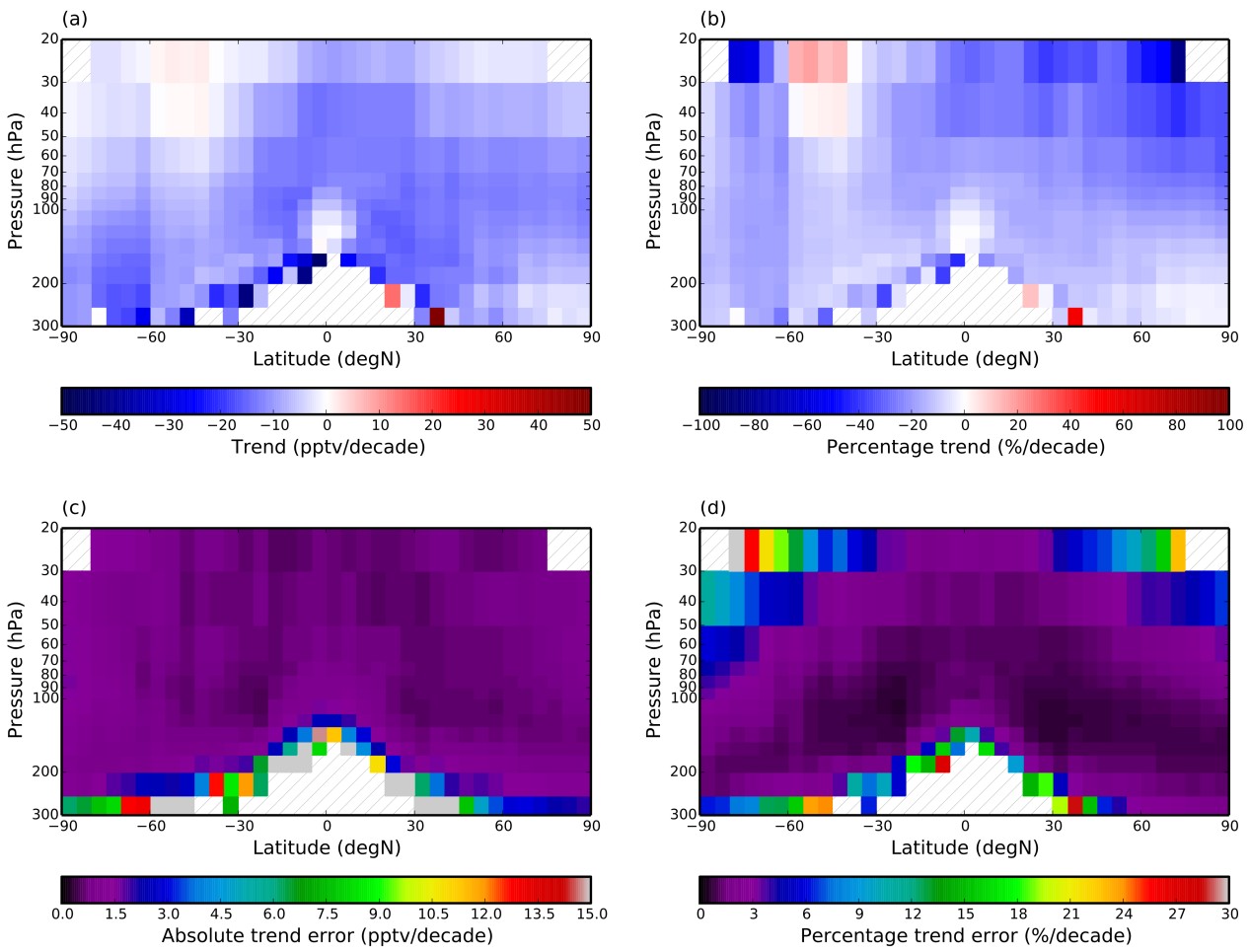

**Figure 11.** CCl$_4$ trends as a function of latitude and pressure. Panel a) absolute trends, b) percentage trends, c) absolute errors, d) percentage errors. Latitudes / pressures with trend error greater than 30% are masked with dashed areas.

Assuming for each latitude bin and pressure level the average CCl$_4$ VMR obtained from the full MIPAS dataset, we also calculated the relative CCl$_4$ trends. They are shown in the panel b) of Fig. 11. The same considerations made for the absolute trends apply also to relative trends. The asymmetry between the NH and the SH is very pronounced, the NH having larger negative relative trends increasing with altitude and reaching 30-35%/decade at 50 hPa. Note however that above 50 hPa they show large variations with both latitude and pressure. These oscillations correspond to extremely small average VMR values that make the relative trend numerically unstable. Panels c) and d) of Fig. 11 show, respectively, the absolute and percentage random errors on the trends. The uncertainties increase above 20 hPa. Large uncertainties are associated to latitude bins and pressure levels for which a relatively small number of measurements is available. For clarity in Fig. 12 we show the ratio

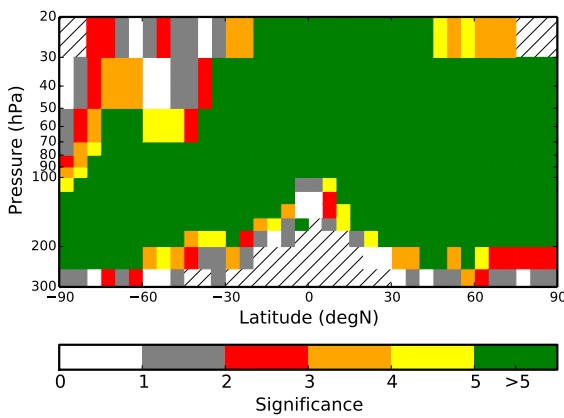

**Figure 12.** Map of the ratio between $CCl_4$ trends and associated random errors.

between $CCl_4$ trends and the related random errors. Latitude bins / pressure levels with ratio values less than 2 are marked with white and grey colors and correspond to trend values that are not significantly different from zero from the statistical point of view. Note, however, that most of the calculated trends are greater than 5 times the related error, and are thus statistically significant. In the maps of Fig.s 11 and 12, values corresponding to errors greater than 30% are masked with dashes. We
consider unreliable any trends with errors greater than this threshold.

As mentioned in Sect. 2.1, an important source of uncertainty could arise from a residual drift of the calibration error, possibly due to neglecting changes in detector non-linearity as the instrument ages. As outlined in Sect. 2.1, however, the worst case scenario for the drift of the calibration error could amount to 1% of the calibration error itself, which in turn, is of the order of 0.4% of each individual retrieved $CCl_4$ VMR profile. Therefore, this error source is negligible compared to the statistical
error shown in panel d) of Fig. 11.

### 5.3  Comparison with $CCl_4$ trends reported in literature

Although measurements acquired at ground stations cannot be directly compared with MIPAS profiles that have a lower altitude limit of 5-6 km, we can still compare tropospheric $CCl_4$ trends derived from MIPAS with trends derived from ground-based measurements. Under the assumption of well-mixed troposphere, we can consider the $CCl_4$ vertical distribution approximately
constant (Chipperfield et al., 2016; Allen et al., 2009). We consider observations provided by two networks that regularly perform long-term, highly accurate near-surface measurements of various tracers, including $CCl_4$: the NOAA/ESRL/HATS (http://www.esrl.noaa.gov/gmd/hats/) and the AGAGE (Simmonds et al., 1998; Prinn et al., 2000, 2016) http://agage.mit.edu/) networks. The NOAA/ESRL/HATS group provides accurate measurements of $CCl_4$ through three different programs: two in situ electron capture detector (ECD) measurement programs and one flask system using gas chromatography with ECD pro-
gram. In this work we use a $CCl_4$ combined dataset, developed by the NOAA to homogenize all of the measurements made by

the different programs (more details at http://www.esrl.noaa.gov/gmd/hats/combined/CCl4.html). All the $CCl_4$ NOAA records are reported on the NOAA-2008 scale. AGAGE measurements used here are obtained using in situ gas chromatography with ECD and reported on the SIO-2005 calibration scale. NOAA and AGAGE in situ measurements at common sites are inter-compared every 6 months for validation purposes.

To compare MIPAS $CCl_4$ trends to those derived from the ground-based measurements of NOAA and AGAGE, we first choose a pressure level belonging to the troposphere, with the following procedure. For each latitude bin ($\lambda$) and MIPAS monthly average profile we identify the tropopause with the pressure level where the monthly average temperature shows its minimum value. We multiply this pressure by 1.6 and find the nearest pressure level ($p_t(\lambda)$) in the fixed pressure grid defined in Sect. 5.1. Using this procedure the selected pressure level is located approximately 3 km below the tropopause. For each

latitude bin and month we then compute the monthly $CCl_4$ average at $p_t(\lambda)$. Finally, for each latitude bin, we calculate the trend at this month- and latitude- dependent tropospheric pressure as explained in Sect. 5.1.

Figure 13 compares the time series of ground-based $CCl_4$ measurements of selected stations (black and orange lines) with MIPAS monthly tropospheric averages (blue dots) in the same latitude bin of the ground station. The two plots refer to ground stations located at tropical (top) and middle (bottom) latitudes. Ground-based measurements do not really show a seasonality,

while MIPAS measurements do. The amplitude of the seasonal variations observed by MIPAS increases with latitude. For tropical latitudes MIPAS OR measurements show a positive bias of approximately 15%. Although not focused on tropical regions, Fig. 8 comparing MIPAS to balloon measurements, already suggests the existence of this bias. At middle latitudes the maximal values of the MIPAS time series roughly match ground measurements. In Fig. 13 we also show the trend values determined on the basis of the plotted measurements. In the examined cases the trends obtained from MIPAS and ground

stations are in very good agreement.

In Table 3 we compare MIPAS tropospheric $CCl_4$ trends with trends derived for the 2002–2012 decade from NOAA/AGAGE stations located in the same latitude band. Some stations produce $CCl_4$ trends in very good agreement with MIPAS. However, in general, and especially in the polar regions, the variability of the tropopause is quite large, thus producing time series of MIPAS monthly averages at $p_t(\lambda)$ that can not be adequately matched by the fitting function defined in Eq. 1. This feature

sometimes generates large residuals in the trend fit and thus large trend errors and/or unrealistic trend values. Despite this difficulty, from the statistical point of view the only trends calculated at the CGO site disagree significantly. We attribute this disagreement to the instabilities occurring in MIPAS data at low altitudes. Indeed, the MIPAS tropospheric trend estimated for the latitude bin $35°/40°$ S (the bin adjacent to the CGO site) is already equal to $-9.16 \pm 2.03$ pptv/decade, i.e. in perfect agreement with the trend calculated from the CGO measurements.

Looking at the literature we found that Brown et al. (2011) estimate the global $CCl_4$ trend from ACE-FTS measurements. The authors consider $CCl_4$ VMR profiles obtained from ACE-FTS in the $30°$ S/$30°$ N latitude belt. They calculate yearly averages of $CCl_4$ VMR in the altitude range from 5 to 17 km and fit the seven 2004-2010 yearly averages with a linear least-squares approach. The resulting trend is $-13.2 \pm 0.9$ pptv/decade. If we average MIPAS trends presented in Sect. 5.2 in the $30°$ S/$30°$ N latitude interval and in the 100–300 hPa pressure range, with a filter discarding trend values with relative error

greater than 30%, we get an average trend of $-12.80 \pm 0.12$ pptv/decade. This value is in very good agreement with the trend

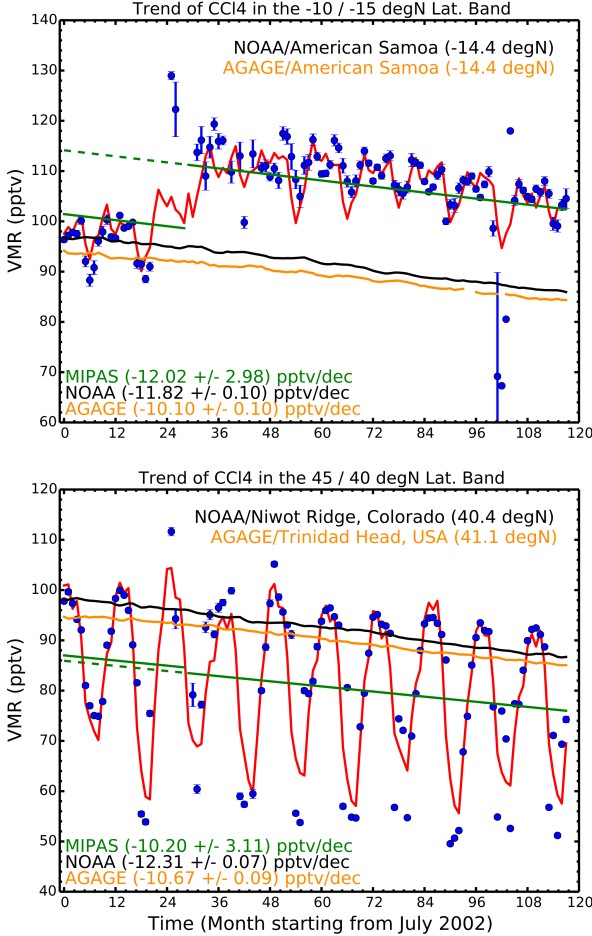

**Figure 13.** Comparison between MIPAS (blue dots) and NOAA/AGAGE (black/orange) $CCl_4$ time series. The two plots refer to ground stations located at tropical (top) and middle (bottom) latitudes. The red curve is the fitting model used to derive the trend from MIPAS data, the green line is the linear part of the model itself. The obtained trend values are also shown in the plots.

determined from ACE-FTS. Note also that, since MIPAS measures atmospheric emission its sampling is finer than that of ACE-FTS both in space and time. With MIPAS it is therefore possible to estimate trends with a better precision.

## 6 Lifetime

In this section, we estimate the stratospheric lifetime of $CCl_4$ according to the tracer-tracer correlation method established by Volk et al. (1997) based on the theoretical framework presented by Plumb and Ko (1992) and Plumb and Zheng (1996). Here we choose $CFC-11$ as the reference tracer ($b$) correlated to $CCl_4$ (tracer $a$). The stratospheric lifetime can be calculated using

| Site Code | Site Name | Latitude (degN) | Network | In-situ trend (pptv/decade) | MIPAS trend (pptv/decade) | MIPAS Lat. Band (degN) |
|---|---|---|---|---|---|---|
| BRW | Barrow, USA | 71.3 | NOAA | −12.7 | −3.2 ± 10.4 | 70/75 |
| MHD | Mace Head, Ireland | 53.3 | AGAGE | −10.1 | −4.7 ± 5.1 | 50/55 |
| THD | Trinidad Head, USA | 41.1 | AGAGE | −10.6 | −10.2 ± 3.1 | 40/45 |
| NWR | Niwot Ridge, USA | 40.4 | NOAA | −12.3 | −10.2 ± 3.1 | 40/45 |
| MLO | Mauna Loa, USA | 19.5 | NOAA | −12.2 | −14.9 ± 2.3 | 15/20 |
| RPB | Ragged Point, Barbados | 13.2 | AGAGE | −10.7 | −12.7 ± 3.6 | 10/15 |
| SMO | Tatuila, American Samoa | −14.4 | NOAA AGAGE | −11.8 −10.1 | −12.0 ± 3.0 | −10/−15 |
| CGO | Cape Grim, Tasmania | −40.7 | AGAGE | −10.2 | −25.9 ± 5.4 | −40/−45 |
| SPO | South Pole, Antartica | −90.0 | NOAA | −11.9 | −7.9 ± 10.6 | −85/−90 |

**Table 3.** For each ground station the table columns show respectively: site code, site name, site latitude, network name, station-related $CCl_4$ trend, tropospheric MIPAS trend, latitudinal band from which MIPAS data were extracted.

the following equation:

$$\frac{\tau_a}{\tau_b} = \frac{\overline{\frac{\sigma_a}{\sigma_b}}}{\frac{d\sigma_a}{d\sigma_b}\big|_{tropopause}} \quad (2)$$

where $\tau_a$ and $\tau_b$ are the stratospheric lifetimes of the two correlated tracers and $\overline{\sigma_a}$, $\overline{\sigma_b}$, $d\sigma_a/d\sigma_b$ are, respectively, the atmospheric VMRs of the two species and the slope of the correlation at the tropopause in steady-state. A major complication that

arises when using Eq. 2 is due the fact that the considered tracers decline in the 2002 - 2012 decade, therefore MIPAS measurements can not be considered to refer to a steady state. Using decadal averages for $\overline{\sigma_a}$ and $\overline{\sigma_b}$ does not actually cause large errors in $\tau_a$, however, replacing the steady state slope with the measured slope $d\chi_a/d\chi_b$ may be a rough approximation (Volk et al., 1997). The difference between the slopes in steady- and transient- states is mainly linked to the tropospheric change rate $\gamma_0$ of the tracers in the considered time period. In order to account for the effect of $\gamma_0$ on $d\sigma_a/d\sigma_b$ we use the following formula

proposed by Volk et al. (1997):

$$\frac{d\sigma_a}{d\sigma_b}\bigg|_{tropopause} = \frac{\frac{d\chi}{d\chi_b}\big|_{tropopause} \cdot \frac{d\chi_b}{d\Gamma}\big|_{\Gamma=0} + \gamma_{0_a}\sigma_{0_a}}{\frac{d\chi_b}{d\Gamma}\big|_{\Gamma=0} + \gamma_{0_b}\sigma_{0_b}} \cdot \frac{1 - 2\gamma_{0_b}\Lambda}{1 - 2\gamma_{0_a}\Lambda}. \tag{3}$$

In this expression $d\chi_b/d\Gamma\big|_{\Gamma=0}$ is the slope of the reference tracer ($b$) with respect to the age of air $\Gamma$ at the tropopause, $\Lambda$ is the width of the atmospheric age spectrum, $\gamma_0$ and $\sigma_0$ are, respectively, the effective linear growth rate and the VMR of the tracers at the tropopause. According to Volk et al. (1997), $\gamma_0$ can be calculated as:

$$\gamma_0 = c - 2\Lambda d \tag{4}$$

where $c$ and $d$ are time-dependent coefficients. At each month ($t$) they are obtained by fitting a 5-years prior time series of monthly VMR averages of the considered tracer at the tropopause level ($\chi_0(t')$) with the following function:

$$\chi_0(t') = \chi_0(t)[1 + c(t' - t) + d(t' - t)^2]. \tag{5}$$

To derive lifetime estimates, as suggested in Brown et al. (2013), we considered only the latitudes in the so-called *surf* zone (Volk et al., 1997), between 30° N/S and 70° N/S. The tropical regions are not suitable to estimate the stratospheric lifetime using the tracer-tracer method due to the intense large-scale upwelling (Plumb and Ko, 1992). Similarly, the polar regions are not suitable for this study due to the intense subsidence, especially during winter (Plumb, 2007). For each month of the MIPAS mission and each 5° latitudinal band between 30° N/S and 70° N/S, we determine the pressure level corresponding to the tropopause, as the level with a minimum in the monthly average temperature profile. For CFC-11 we assume a lifetime $\tau_b = 52(43 - 67)$ years (SPARC, 2013). To determine the coefficients $c$ and $d$ appearing in Eq. 5, at each MIPAS measurement month $t$ we fit a time series of HATS (http://www.esrl.noaa.gov/gmd/hats/) $CCl_4$ and CFC-11 global monthly averages. Each time series extends back in time for 5 years, starting from the month $t$. The calculation is then repeated for each month of the MIPAS mission, from April 2002 to July 2012. For the estimation of lifetimes limited to NH and SH we used respectively NH and SH HATS monthly means instead of global monthly mean. We then used the coefficients $c$ and $d$ to calculate the effective linear growth rate $\gamma_0$ via Eq. 4, assuming $\Lambda =1.25$ years as suggested in Volk et al. (1997) and in Laube et al. (2013).

To estimate the slope of CFC-11 with respect to the age of air at the tropopause we used an analysis of air samples acquired on board *Geophysica* aircraft (Laube et al., 2013). The analysis produces a $d\chi_b/d\Gamma\big|_{\Gamma=0}$ value of $-20.6 \pm 4.6 \; ppt \; yr^{-1}$ for 2010. We calculated the slope for other years by scaling the 2010 value according the relative change of the yearly $\gamma_0$ average. For Eq. 3 we used an average of the $\gamma_0$ values obtained in the whole MIPAS mission period.

We determined the slope of the correlation at the tropopause $d\chi_a/d\chi_b\big|_{tropopause}$ according to the method suggested by Brown et al. (2013). We considered only the VMR monthly means of CFC-11 and $CCl_4$ at the SPARC pressure levels (see Sect. 5.1) above the tropopause. First of all, the mean correlation curve has been created calculating the mean of the $CCl_4$ data within 2 pptv of CFC-11 wide windows. The slope of the data has been calculated using a linear least squared fit within a moving window of 80 pptv of CFC-11. After the calculation, the moving window would be shifted forward by 5 pptv and the slope would be calculated again. The procedure was repeated for each 5 degrees latitudinal band. As suggested in Brown

et al. (2013) only CFC-11 VMRs greater than 120 pptv are considered. This approach makes us confident that the calculated slope is not affected by VMR values arising from the upper stratosphere. The remaining data were fitted using a second degree polynomial to calculate the value of the slope at the tropopause.

We calculated the VMR at the tropopause ($\sigma_0$) by averaging all the VMR monthly averages at the tropopause pressure level. The monthly means are then weighted using the corresponding atmospheric pressure. The atmospheric VMR ($\overline{\sigma}$) is calculated averaging the VMR monthly averages weighted with atmospheric pressure, in the pressure range between 200 and 20 hPa. The calculation of $\sigma_0$ and $\overline{\sigma}$ of $CCl_4$ and CFC-11 is carried-out separately for each latitudinal band, yielding a $CCl_4$ global average lifetime of 47(39 - 61) years, a lifetime of 49(40 - 63) years in the NH, and 46(38 - 60) years in the SH. We calculated the $CCl_4$ lifetime confidence interval by mapping through the calculations the CFC-11 lifetime confidence interval (see SPARC (2013, 2016) for more details). We also evaluated the impact of other possible error sources using a perturbative approach. We found that a 10% bias in the $CCl_4$ VMR retrieved from MIPAS (see Sect. 4) would cause an error of the order of $3-4\%$ in the $CCl_4$ lifetime. An uncertainty of $\pm 4.6\ ppt\ yr^{-1}$ in $d\chi_b/d\Gamma|_{\Gamma=0}$ would cause an error smaller than $4\%$ in the $CCl_4$ lifetime. These contributions are by far smaller than the error implied by the uncertainty in the CFC-11 lifetime.

Our $CCl_4$ lifetime estimations are consistent with the most recent literature that suggests an atmospheric lifetime of 44(36 - 58) years (SPARC, 2013, 2016). Several older studies report atmospheric $CCl_4$ lifetimes between 30 and 50 years (Singh et al., 1976; Simmonds et al., 1988; Montzka et al., 1999; World Meteorological Organization (WMO), 1999; Allen et al., 2009). Brown et al. (2013) studied the stratospheric lifetime of several species (including CFC-11 and $CCl_4$) using ACE-FTS measurements. Using a CFC-11 lifetime of 45±7 (World Meteorological Organization (WMO), 2011) they calculated a $CCl_4$ global lifetime of 35±11 years. The difference with our results is explained taking into account the different reference CFC-11 lifetimes used: using the same CFC-11 lifetime (World Meteorological Organization (WMO), 2011) we would obtain a $CCl_4$ lifetime of 41±6 years. Brown et al. (2013) report also very different lifetimes in the two hemispheres (41±9 years in the NH and 21±6 years in the SH) but they are not able to provide a solid justification for this finding. Again, the differences with our results are partially explained with the different CFC-11 lifetime considered (using the same CFC-11 lifetime (World Meteorological Organization (WMO), 2011) we would obtain a $CCl_4$ lifetime of 42±7 years in the NH and 40±6 years in the SH) but the choice of different reference lifetimes does not explain the hemispheric asymmetry reported in Brown et al. (2013).

## 7 Conclusions

The ESA Version 7 processor has been used to determine for the first time the $CCl_4$ VMR global distribution in the UTLS using MIPAS measurements. The MIPAS $CCl_4$ observations cover the altitude range from 6 to 27 km and, having been obtained from emission measurements, provide a global coverage. The zonal means of $CCl_4$ VMR show features typical of long-lived species of anthropogenic origin that are destroyed primarily in the stratosphere by photolysis. The highest VMR values are found in the troposphere, and VMR monotonically decreases with increasing altitude in the stratosphere. In the lower stratosphere, the largest values are observed between 30°S and 30°N due to the intense updraft that occurs in the tropical region. The $CCl_4$

global distribution shows also a seasonal variability. This seasonality is more evident in the polar regions due to $CCl_4$-poor mesospheric air subsidence induced by the polar vortex.

We calculated inter-hemispheric VMR differences in the UTLS as a function of pressure and latitude using MIPAS average $CCl_4$ profiles. At high latitudes, the asymmetry likely stems from the fact that the polar vortex in the Antarctic is systematically stronger, more stable, and of longer duration than the Arctic polar vortex. At mid-latitudes, NH and SH seasons are more symmetrical and the $CCl_4$ mean differences between the two hemispheres are probably caused by the larger $CCl_4$ emissions in the NH (SPARC, 2016; Liang et al., 2014). The weighted mean of NH-SH $CCl_4$ differences in the lowermost pressure levels sounded by MIPAS is consistent with the IHG value reported by Liang et al. (2014).

We compared MIPAS $CCl_4$ profiles to profiles derived from the balloon version of MIPAS (MIPAS-B) and from the solar occultation ACE-FTS instrument. While MIPAS-B inter-comparison covers both FR and OR mission phases at selected latitudes, ACE inter-comparison covers the OR phase, globally, for latitudes larger than 45 degrees. In general, MIPAS/ENVISAT measurements are within 10% of both instruments for pressures between 100 and 40 hPa. A positive bias is found mainly in tropical regions at very low altitudes for OR measurements. In the latitude band $50°S-70°S$, MIPAS shows a larger negative bias with respect to ACE-FTS, but this bias seems to reduce when compared with the upcoming version of ACE-FTS products. For pressures smaller than 40 hPa, MIPAS/ENVISAT $CCl_4$ values are between MIPAS-B and ACE-FTS.

We used the $CCl_4$ measurements to estimate for the first time the $CCl_4$ trends as a function of both latitude and pressure, including the photolytic loss region (70-20 hPa). Negative trends ($-10/-15$ pptv/decade, $-10/-30$ %/decade) are observed at all latitudes in the UTLS region, with the exception of slightly positive values ($5/10$ pptv/decade, $15/20$ %/decade) for a limited region at Southern mid-latitudes between 50 and 10 hPa. We attribute positive stratospheric trend to the less effective mixing mechanisms in the stratosphere as compared to the troposphere at these latitudes. In general, $CCl_4$ VMR values exhibit a smaller decline rate for the SH than the NH. The magnitude of the negative trend increases with altitude, more strongly in the NH, reaching values of 30-35%/decade at 50 hPa, close to the lifetime limited rate. The hemispheric asymmetry of the trend is probably related to the asymmetry in the general circulation of the atmosphere.

An approach based on tracer-tracer linear correlations was used to estimate $CCl_4$ atmospheric lifetime in the lower stratosphere. The calculation provides a global average lifetime of 47(39 - 61) years considering CFC-11 as reference tracer. These results are consistent with the most recent literature results of 44(36 - 58) years (SPARC, 2013, 2016). We also computed the $CCl_4$ lifetime separately for the two hemispheres, obtaining 49(40 - 63) years for the NH and 46(38 - 60) years for the SH.

## 8 Data availability

MIPAS ESA Level 2 products Version 7 can be obtained via https://earth.esa.int/web/guest/data-access (registration required). Trend values and related errors used to build the maps of Fig. 4 are available upon request to the authors.

**Aknowledgements**

MIPAS ESA V7 data were processed in the frame of Contract no. 21719/08/I-OL funded by ESA. We thank AGAGE leaders R. Prinn, R.Weiss, P. Krummel and S. O'Doherty for providing AGAGE data. AGAGE is supported principally by NASA (USA) grants to MIT and Scripps Institution of Oceanography, and also by DECC (UK) and NOAA (USA) grants to Bristol University and by CSIRO and BoM (Australia). We thank NOAA Climate Program Office for the support and for the NOAA/ESRL/HATS $CCl_4$ data. Funding for the Atmospheric Chemistry Experiment was supplied primarily by the Canadian Space Agency.

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
