# Peer review of "CCl4 distribution derived from MIPAS ESA V7 data: intercomparisons, trend and lifetime estimation"

_Atmospheric Chemistry and Physics, 2016_

## Referee Comment (RC1) · Anonymous Referee #3 · 27 Mar 2017

Review of

CCl4 distribution derived from MIPAS ESA V7 data: validation, trend and lifetime estimation

Valeri et al

Overview

The paper presents the results of an analysis of the new MIPAS CCl4 product from the ESA processor. While opportunities for validation are limited the authors do exploit one of the strengths of MIPAS, which is a 10-year globally sampled dataset to draw conclusions on interhemispheric variation and trends. On the whole, the paper is a clearly-written and convincing and I have no major criticisms.

General comments

a) While there is a convincing trend (matching the ground stations) it would have been useful to apply the same trend analysis to a different molecule retrieved with the same algorithm (eg N2O?) which has no expected trend. This would help quantify the contribution of any calibration drift.

b) Of all the time-series fit parameters, it would have been helpful to indicate which ones were actually significant: the trend, constant and annual cycles are obvious from Fig 10 but what effect do the other terms have? Were they really needed?

c) Comparison with ground stations: is the assumption here that the CCl4 profile is expected to be constant with altitude all the way through the troposphere? It would have been helpful to show at least a modelled CCl4 profile to support this. However, the fact that the MIPAS data have a seasonal cycle while the ground station data do not suggests that these must be different air masses, in which case there is presumably also some age difference between the air sampled by MIPAS and the surface air which could explain some of the bias.

d) Given the data available, it is possible to calculate a *total* atmospheric content of CCl4, at least the partial column above some pressure surface, and provide the trend of this with time. This would be a much easier quantity for simple comparison with models or other satellite instruments without having to match details of pressure levels or latitude bands, also for stratospheric chlorine budgets.

Minor comments

P2 L5: It is not clear from the text whether CCl4 is an entirely anthropogenic gas or whether there is also some (small?) natural source.

P4 L19: If you mention 'oversampling the limb' you should explain what the size of the field-of-view is.

P4 L21: 8 rows for the FR AK, but only 7 for OR.

P7 Much of the text here us unnecessary as it is already in the Fig 3 caption.

P9 Presumably the effect is larger in the antarctic due to the stronger, more stable polar vortex?

P10 L6: Since the ocean is the major surface sink, and there is more ocean in the southern hemisphere, wouldn't an IHG be expected even in the absence of continued emissions?

P11 L5/Fig 6: since Fig 6 is effectively an annual average its difficult to argue which components are persistent and which are seasonal. Perhaps there's an alternative way of plotting the data to highlight the seasonal differences (eg shift the s.hemisphere data by 6 months before subtracting?)

P11 L14: I can understand why balloon instruments might have better signal/noise than satellite instruments since they can effectively take many scans of the same atmosphere, but I don't understand what is instrinsic to the balloon measurement that gives it high vertical resolution compared to satellites. Indeed the 1.5km spacing of MIPAS-B seems comparable to MIPAS.

P15 L12: Given that CCl4 is a relatively long-lived gas with no diurnal variation, and that both MIPAS and ACE-FTS obtain relatively uniform sampling in longitude, I wonder why you didn't simply compare zonal means of both datasets (interpolating MIPAS to the approrpriate latitude for ACE-FTS each day) rather than look for profile-by-profile coincidences which could contain a latitude bias or end up just selecting MIPAS ascending or descending node observations (with the associated GRAD error).

P15 L15: Again much of the text repeats what is in the figure caption, although it takes a while before explaining what I really wanted to know, which is the distinction between 'standard deviation of the mean' and 'standard deviation of the differences'. The former is just the latter divided by root(N), is that right?

P17 Eq(1): I agree with the approach but the term 'offset parameters' confused me -

offset relative to what? Perhaps just 'constant parameters'.

Typographic/grammatical comments

P1 L1: no need for capital C in 'Carbon tetrachloride'

P1 L12: 20-50 rather than 20/50 if this indicates a range of latitudes rather than a particular pair of latitudes P3 L9: Similarly.

P2 L33: Suggest 'limits' rather than 'edges'.

P3 L14: 'where' rather than 'were'

P15 L6: Suggest 'extends' rather than 'goes'

Fig 5: some vertical lines at the year boundaries would be helpful.

Fig 6: 'degN' for the latitude axis should presumably just be 'deg' here.

---

## Referee Comment (RC2) · Anonymous Referee #4 · 30 Mar 2017

I think that this is a useful and important paper which is well suited to publication in ACP. There has been a lot of interest in atmospheric CCl4 because of an apparent 'budget gap'. An important sink term for CCl4 is atmospheric loss and to evaluate our understanding of that process profile observations into the stratosphere are required. This paper presents such data from the MIPAS instrument which has the benefit of a lot of observations to average over.

I think that the paper can be published subject to my comments below.

Main points

1) Throughout the paper could benefit from a thorough proof-reading. There are some simple spelling errors that any spell checker should find. There are also some other sentences where the English is poor. The quality does vary through the paper (e.g. the

abstract in particular had many typos). I have mentioned some below, but in addition the paper needs careful proof reading.

2) Stratospheric trends. A number of recent papers have shown that the trends in stratospheric trace gases are affected by variability in the stratospheric circulation. This has been shown for a number of halogen source gases and the complementary degradation products such as HCl and HF. This is bound to be playing a role in the stratospheric trends shown in Figure 11 and will be at least part of the explanation of why the trend does not simply follow the tropospheric trend (with a lag). I know there is mention in the Conclusions (page 26 line 5) but more should be added near Figure 11. It is a case of adding in some mention of past work. Examples to cite are:

Harrison, J.J., M.P. Chipperfield, C.D. Boone, S.S. Dhomse, P.F. Bernath, L. Froidevaux, J. Anderson and J.M. Russell, Satellite observations of stratospheric hydrogen fluoride and comparisons with SLIMCAT calculations, Atmos. Chem. Phys., 16, 10,501-10,519, doi:10.5194/acp-16-710501-2016, 2016.

Mahieu, E., M.P. Chipperfield, J. Notholt, T. Reddmann, J. Anderson, P.F. Bernath, T. Blumenstock, M.T. Coffey, S. Dhomse, W. Feng, B. Franco, L. Froidevaux, D.W.T. Griffith, J. Hannigan, F. Hase, R. Hossaini, N.B. Jones, I. Morino, I. Murata, H. Nakajima, M. Palm, C. Paton-Walsh, J.M. Russell, M. Schneider, C. Servais, D. Smale and K.A. Walker, Recent northern hemisphere hydrogen chloride increase due to atmospheric circulation change, Nature, 515, 104-107, doi:10.1038/nature13857, 2014.

Ploeger, F., Riese, M., Haenel, F., Konopka, P., Müller, R., and Stiller, G.: Variability of stratospheric mean age of air and of the local effects of residual circulation and eddy mixing, J. Geophys. Res.-Atmos., 120, 716–733, doi:10.1002/2014JD022468, 2015.

3) Figure 6 does not make sense to me. Normally the N-S IHG is presented based on an average over the two hemispheres. How is Figure 6 constructed? Is it the difference between corresponding latitudes (e.g. 80S minus 80N)? That does not make sense as the high latitudes get more and more distant from the other hemisphere so the scope for

differences is much larger. There is also less mass at high latitudes so the differences are not so important in a budget sense. I think that this figure is flawed and should be removed.

Minor Points

Abstract line 1. Change 'strong' to 'potent'?

Page 1. Line 4. Typo: mystery.

Page 1. Line 6. Typo: photolytic.

Page 1. Line 9. Typo: anthropogenic.

Page 1. Line 12. 'proves' is too strong. Could change to 'gives confidence in' (or similar).

Page 1. Line 16. Change scan to scans?

Page 2. Line 1. ODP is ozone *depletion* potential.

Page 2. Line 6. Typo: hydrofluorocarbons.

Page 3. Line 4. Typo: where.

Page 3. Line 5. Here you could cite a recent paper on modelling the CCl4 budget using the latest lifetime data and limited ACE CCl4 data to evaluate the model stratosphere. The availability of more stratospheric data would help constrain such model studies.

Chipperfield, M.P., Q. Liang, M. Rigby, R. Hossaini, S.A. Montzka, S. Dhomse, W. Feng, R.G. Prinn, R.F. Weiss, C.M. Harth, P.K. Salameh, J. Muhle, S. O'Doherty, D. Young, P.G. Simmonds, P.B. Krummel, P.J. Fraser, L.P. Steele, J.D. Happell, R.C. Rhew, J. Butler, S.A. Yvon-Lewis, B. Hall, D. Nance, F. Moore, B.R. Miller, J.W. Elkins, J.J. Harrison, C.D. Boone17, E.L. Atlas and E. Mahieu, Model sensitivity studies of the decrease in atmospheric carbon tetrachloride, Atmos. Chem. Phys., 16, 15,741-15,754, doi:10.5194/acp-16-15741-2016, 2016

Page 3. Line 21. 'operation' (singular).

Page 3. Line 32. Change to 'allowing the study of the evolution of atmospheric composition in great detail'.

Page 4. Table 1. Spell out MW in the caption.

Page 4. Line 12. Change to 'includes only one out of every two'.

Page 5. Figure 1 caption. Specificy 'coloured solid lines'.

Page 5. Line 4. 'Apart from the "NLGAIN"...'

Page 6. Line 7. Do these errors 'cancel out' exactly? If not you should say something like 'largely cancel out. . .'.

Page 7. Line 2. Typos: '. . ..type of error, therefore, has no impact on the trend calculation'.

Page 7. Line 19. 'We do not show..'

Page 7. Lines 25-29. These lines are not clear to me. I think it is the use of the word 'compatible'. You should look into rephrasing this.

Page 8. Line 5. 'continuing for inertia'. This does not make sense and needs to be rephrased.

Page 8. Line 11. 'hemispheres' (small h).

Page 8. Line 12. 'troposphere' must be a typo? At 130 hPa high latitudes will be in the stratosphere.

Page 8. Line 14. Change 'notice' to 'note'.

Page 9. Line 7. Typo: 'transport'.

Page 9. Line 8. 'justify' is the wrong word. Use 'explain'?

Page 12. Line 7. 'Further to'. 'simultaneously'.

Page 12. Line 20. 'incompatible'.

Page 13. Figure 7 (and 8). The caption should explain the red numbers on the left panel.

---

## Referee Comment (RC3) · Anonymous Referee #2 · 15 Apr 2017

This manuscript describes the retrieval and interpretation of a near-global data set of the atmospheric trace gas carbon tetra chloride (CCl4) from the MIPAS satellite instrument as obtained between 2002 and 2012. I consider the manuscript to be publishable in ACP after the points outlined below have been addressed, in particular the ones regarding the amount of quantitative information and the lifetime estimates. In addition I urge the authors to reconsider the excessive use of abbreviations which is limiting readability.

p1 l3. The recent SPARC report with that name should be credited here. Given that it was a very recent and international effort on CCl4 I find that report has been cited and used very little throughout the manuscript.

p1 l12. This statement and evidence for it is nowhere to be found in the manuscript.

p1 l12-14. I disagree. This good agreement only proves that the remote sounders are producing similar results, but it is not a validation.

p1 l15-20. I would strongly recommend some more quantitative information in this section. What are the actual trends, the lowest altitudes sounded by MIPAS, and the comparability of the mixing ratios and trends, including uncertainties? Also, how do the authors explain the positive trend in the Southern mid latitudes?

p4 l20-25. Most of that section should be moved to the caption of the figure. In fact most figure captions in the manuscript need more explanation of what is shown.

p8 l 11 & 14. There are still quite a few minor English language problems in this manuscript, two examples here are "CCl4-poor" and "in the South Pole".

p8 l11. If there is a seasonal effect it is not obvious from figure 4. Can the authors quantify this seasonality, also to prove that it is indeed statistically significant? A similarly quantitative approach would help in other parts of the manuscript too, e.g. the earlier statements on latitudinal and altitudinal gradients.

p9 figure 4 caption. "May 20117"

p11 l9-11. This is not correct. Numerous aircraft and balloon campaigns have measured CCl4 with alternative in situ techniques. Please see e.g. Volk et al., 1997 and the many papers that cite it, as well as the FTIR total column measurements from the Jungfraujoch station.

p16 l4-6. This is exactly where alternative validation methods could help.

p23 l10. A "kind of global CCl4 trend"?

p24 l6. The smaller trend error does not take into account the biases, though.

p24 section 6. This section needs some additional work. The methodology (equation 2) is not used correctly as Plumb and Ko (1992) clearly state that a) it should only be applied to two species in steady state and b) the slope needs to be determined exactly

at the tropopause. Moreover the method was improved by Volk et al.,1997 and Brown et al., 2013 to e.g. correct for tropospheric trends and derive steady-state lifetimes. A second problem with the lifetime estimate presented here is that it is highly dependent on uncertainties and potential biases of the trace gases involved, i.e. CCl4, CFC-11 and CFC-12. Can the authors present evidence that these uncertainties and biases have been taken into account for the determination of the lifetime and its uncertainties?

---

## Referee Comment (RC4) · Anonymous Referee #1 · 26 Apr 2017

**GENERAL COMMENTS**

The paper describes the CCl$_4$ VMRs as derived from the MIPAS instrument using the ESA V7 data set. This data set is validated against independent measurements and used to determined trends and lifetime of this trace gas, which are consistent with other recent estimates.

The paper is very well written and its scope fits well into AMT. However, there are some questions that should be clarified before publication.

**SPECIFIC COMMENTS**

*page 9, Figure 4:*
The lowermost values at several latitude bands exhibit drastically increased VMRs

(compared to values above and beside). Are these values typical for deep tropospheric VMRs or could, e.g., (undetected) thin clouds or stray light have affected the measurements? Would filtering these extreme values affect the trend analysis in a positive or negative way?

*page 12, line 21ff:*
It is not fully clear what the discussed quantities of Fig. 6 and 7 are. I assume that the blue curves are simply the sum of the errors of the two individual instruments? Or was the precision as a random error summed in the square? This is rather elaborately described for the ACE-FTS instrument following this section but missing here. Further I do not fully understand the distinction between the standard deviation "sd" of the differences and the error bars on the mean. How was the standard deviation of the mean computed? By dividing the standard deviation of the differences by the square root of measurements? Or was a jackknife-like algorithm employed? Further, was the standard deviation of the differences computed with an assumed mean of zero? Otherwise, shouldn't those standard deviations be plotted relative to the mean instead of the zero line?

*page 23, Table 3:*
What are the pressure levels chosen for the MIPAS data? At -45 degree latitude, the significance of the data is reduced at 200 hPa and below. I am not sure that I can identify the box with a trend of 25 pptv and an error of only 5 pptv between -40 and -45. The values are difficult to determine using the continuous colour scale, but the lowest box in this grid seems to have a value of -15+-5.

**MINOR REMARKS**

*page 8, line 6:*
Space after "Sect." is missing.

*page 12, line 21ff:*
How much of the difference can be attributed to the different level 2 algorithms (e.g.

employed micro windows and spectral databases)?

*page 17, line 17:*
What is the reasoning behind the specific value of 1.6? Obviously one is looking for a grid point being "always" in the troposphere with a sufficient distance from the stratosphere as to not be influenced by its value (more than 1.5km distance?) but as high as possible as the significance drops with altitude. I would expect that for many latitude bands no significant value would be available.

*page 17, line 28f:*
What is the specific reasoning for including this specific set of oscillation periods and how significant are the determined factors $c_i$ and $d_i$?

———————————————————

---

## Author Comment (AC3) · 8 Jun 2017

The comment was uploaded in the form of a pdf file that can be downloaded from the link below.

Please also note the supplement to this comment:
http://www.atmos-chem-phys-discuss.net/acp-2016-1163/acp-2016-1163-AC3-supplement.pdf

---

## Author Response (AR1)

**Subject**: submission of a revised version of the manuscript acp-2016-1163

Dear Editor,

please find enclosed our point-by-point replies to referee comments, along with a description of the changes implemented in the revised manuscript. We also include a marked-up manuscript version showing the changes made.

Thank-you very much for your kind support.

Best regards,

The authors

We would like to thank the reviewer for useful comments. In the following we answer the specific comments (included in "**boldface**" for clarity) and, whenever required, we describe the related changes implemented in the revised manuscript.

**Anonymous Referee #1**

**GENERAL COMMENTS:**

**The paper describes the CCl4 VMRs as derived from the MIPAS instrument using the ESA V7 data set. This data set is validated against independent measurements and used to determine trends and lifetime of this trace gas, which are consistent with other recent estimates. The paper is very well written and its scope fits well into AMT. However, there are some questions that should be clarified before publication.**

**SPECIFIC COMMENTS**

*page 9, Figure 4:* **The lowermost values at several latitude bands exhibit drastically increased VMRs (compared to values above and beside). Are these values typical for deep tropospheric VMRs or could, e.g., (undetected) thin clouds or stray light have affected the measurements? Would filtering these extreme values affect the trend analysis in a positive or negative way?**

The VMRs at lowermost pressure levels can be affected by the presence of thin clouds causing extreme values and more scattered monthly mean time series. Nevertheless the fit is able to manage these values. In these cases the quality of the fit is poor and the chi-squared is greater. Large chi-squared values imply large uncertainties on the trend and small significativity. Fig's. 11 and 12 clearly show this effect: at lowermost pressure levels the uncertainty is larger and the significativity is smaller as compared to the values at higher altitudes.

*page 12, line 21ff:* **It is not fully clear what the discussed quantities of Fig. 6 and 7 are. I assume that the blue curves are simply the sum of the errors of the two**

**individual instruments? Or was the precision as a random error summed in the square? This is rather elaborately described for the ACE-FTS instrument following this section but missing here. Further I do not fully understand the distinction between the standard deviation "sd" of the differences and the error bars on the mean. How was the standard deviation of the mean computed? By dividing the standard deviation of the differences by the square root of measurements? Or was a jackknife-like algorithm employed? Further, was the standard deviation of the differences computed with an assumed mean of zero? Otherwise, shouldn't those standard deviations be plotted relative to the mean instead of the zero line?**

In the revised paper we included the following additional description in the caption of Fig. 7: "The plots show mean absolute and relative VMR differences of trajectory match collocations (red numbers) between both MIPAS sensors (red solid line) including standard deviation of the difference (red dotted lines) and standard error of the mean (plotted as error bars). Precision (blue dotted lines), systematic (blue dash-dotted lines) and total (blue dashed lines) mean combined errors calculated according to the error summation $(err_{MIPAS-E}^2 + err_{MIPAS-B}^2)^{0.5}$ are displayed, too. For further details on the error calculation, see Wetzel et al. (2013)."

The standard deviation of the differences is not computed with a zero mean but with the actual mean. Anyhow, it makes sense to plot it relative to the zero line such that it is directly comparable to the precision.

Caption Figure 8: Same as Figure 7 but for the OR part of the MIPAS mission.

We also added the following reference:

Wetzel, G., Oelhaf, H., Berthet, G., Bracher, A., Cornacchia, C., Feist, D. G., Fischer, H., Fix, A., Iarlori, M., Kleinert, A., Lengel, A., Milz, M., Mona, L., Müller, S. C., Ovarlez, J., Pappalardo, G., Piccolo, C., Raspollini, P., Renard, J.-B., Rizi, V., Rohs, S., Schiller, C., Stiller, G., Weber, M., and Zhang, G.: Validation of MIPAS-ENVISAT $H_2O$ operational data collected between July 2002 and March 2004, Atmos. Chem. Phys., 13, 5791-5811, doi:10.5194/acp-13-5791-2013, 2013.

*page 23, Table 3:* **What are the pressure levels chosen for the MIPAS data? At -45 degree latitude, the significance of the data is reduced at 200 hPa and below. I am not sure that I can identify the box with a trend of 25 pptv and an error of only 5 pptv between -40 and -45. The values are difficult to determine using the continuous colour scale, but the lowest box in this grid seems to have a value of -15+-5.**

In Table 3 MIPAS trends are calculated at variable pressure levels (as explained in the text, page 21 lines 14-19). Therefore the trend values reported in table 3 are not directly comparable to those shown in Fig. 11.

**MINOR REMARKS**

*page 8, line 6:* **Space after "Sect." is missing.**

Done.

*page 12, line 21ff:* **How much of the difference can be attributed to the different level 2 algorithms (e.g. employed micro windows and spectral databases)?**

As written on page 12 line 9, CCl4 cross sections used by MIPAS-B are the same as the ones used by MIPAS/ESA version 7 retrievals. However, the selection of microwindows used for the retrievals of both sensors is different (as mentioned on page 12, line 7 and in Table 1). This might explain at least part of the differences where CCl4 amounts are low (above about 24 km). We added a corresponding sentence in the text.

*page 17, line 17:* **What is the reasoning behind the specific value of 1.6? Obviously one is looking for a grid point being "always" in the troposphere with a sufficient distance from the stratosphere as to not be influenced by its value (more than 1.5km distance?) but as high as possible as the significance drops with altitude. I would expect that for many latitude bands no significant value would be available.**

We guess the reviewer refers to page 21, line 17. As mentioned by the reviewer the major complication in this procedure is to find the 'correct' pressure level ("… in the troposphere with a sufficient distance from the stratosphere …  but as high as possible as

the significance drops with altitude. …"). As mentioned in the paper, we select this pressure level as follows: we identify the pressure at the tropopause and we choose the pressure-grid level closest to the tropopause pressure increased by 60%. The aim of this procedure is to include in the trend-calculation analysis only VMR values relating to a pressure level located about 3 km below the tropopause. Unfortunately, whenever the tropopause is very low (i.e. at high latitudes) the significance of the derived trend decreases, due to the same problems identified by the reviewer in the above comment referring to page 9, fig.4. This effect impacts the trend fit and consequently produces a large error of the trend at high latitudes, as evident from MIPAS trend errors reported in Table 3.

In the revised paper we have rephrased the sentence. "We multiply this pressure by 1.6 and find the nearest pressure level $(p_t(\lambda))$ in the fixed pressure grid defined in Sect. 5.1." → "We multiply this pressure by 1.6 and find the nearest pressure level $(pt(\lambda))$ in the fixed pressure grid defined in Sect. 5.1. Using this procedure the selected pressure level is located approximately 3 km below the tropopause pressure level".

*page 17, line 28f:* **What is the specific reasoning for including this specific set of oscillation periods and how significant are the determined factors Ci and Di?**

This set of oscillation periods has been previously used in several recent papers (Kellmann et al., 2012; Eckert et al., 2014; Haenel et al., 2015). As explained by Haenel et al. (2015): "The period of the first two sine and cosine functions is 12 and 6 months respectively, representing the seasonal and the semiannual cycle. The other six terms have period lengths of 3, 4, 8, 9, 18 and 24 months and describe deviations of the temporal variation from a pure sine or cosine wave. Fitting sine and cosine of the same period length accounts for a possible phase shift of the oscillation."

To avoid repetition we have added a reference to Haenel et al. (2015) near the description of the oscillation periods used in this work.

Haenel, F. J., Stiller, G. P., von Clarmann, T., Funke, B., Eckert, E., Glatthor, N., Grabowski, U., Kellmann, S., Kiefer, M., Linden, A., and Reddmann, T.: Reassessment of MIPAS age of air trends and variability, Atmos. Chem. Phys., 15, 13161-13176, doi:10.5194/acp-15-13161-2015, 2015.

We would like to thank the reviewer for useful comments. In the following we answer the specific comments (included in "**boldface**" for clarity) and, whenever required, we describe the related changes implemented in the revised manuscript.

**Anonymous Referee #2**

**This manuscript describes the retrieval and interpretation of a near-global data set of the atmospheric trace gas carbon tetra chloride (CCl4) from the MIPAS satellite instrument as obtained between 2002 and 2012. I consider the manuscript to be publishable in ACP after the points outlined below have been addressed, in particular the ones regarding the amount of quantitative information and the lifetime estimates. In addition I urge the authors to reconsider the excessive use of abbreviations which is limiting readability.**

**p1 l3. The recent SPARC report with that name should be credited here. Given that it was a very recent and international effort on CCl4 I find that report has been cited and used very little throughout the manuscript.**

We agree with the reviewer and we have added more references to the recent SPARC report throughout the paper. At the same time we think that the abstract of a paper must be a stand-alone, so we preferred not to add the citation directly in the abstract.

**p1 l12. This statement and evidence for it is nowhere to be found in the manuscript.**

In the discussion of Fig. 6, we attributed the North Hemisphere – South Hemisphere (NH-SH) differences at middle latitudes to larger emissions in the NH, however the evidence is not directly related to the results of this paper. For this reason the sentence "In the troposphere, the largest values are observed at latitudes of major industrial countries (20$^{\circ}$/50$^{\circ}$N)." has been removed.

**p1 l12-14. I disagree. This good agreement only proves that the remote sounders are producing similar results, but it is not a validation.**

We fully agree with the reviewer, we also decided to modify the title of the paper from "CCl4 distribution derived from MIPAS ESA V7 data: validation, trend and lifetime estimation" to "CCl4 distribution derived from MIPAS ESA V7 data: inter-comparisons,

trend and lifetime estimation"

The sentence has been rephrased. "The good agreement we find between MIPAS CCl4 and independent measurements from other satellite and balloon-borne remote sounders proves the reliability of the MIPAS dataset." → "MIPAS CCl4 measurements have been compared with independent measurements from other satellite and balloon-borne remote sounders showing a good agreement between the different datasets."

**p1 l15-20. I would strongly recommend some more quantitative information in this section. What are the actual trends, the lowest altitudes sounded by MIPAS, and the comparability of the mixing ratios and trends, including uncertainties? Also, how do the authors explain the positive trend in the Southern mid latitudes?**

In the revised paper we tried to include additional numbers in the abstract, even if it is not always possible to summarize with a few numbers the information contained in the maps. Throughout the paper we report plots/maps that quantify with great details the variability of the results as a function of latitude and / or height. One of the key points of this work is to exploit MIPAS measurement capabilities to highlight the variability of trends as a function of latitude and pressure / altitude. In many cases, due to complexity of the studied phenomenon, the results can't be summarized in a few numbers. Plots and maps represent a more comprehensive picture of the studied processes. As far as numbers are concerned, as explained in the "Data availability" Section of the paper, these are freely available upon request to the authors.

About the positive trend in the Southern mid-latitudes and its possible explanation we improved the discussion in Sect. 5.2 by adding some comments and references to recent works suggested by another reviewer.

**p4 l20-25. Most of that section should be moved to the caption of the figure. In fact most figure captions in the manuscript need more explanation of what is shown.**

Done. The new caption of Fig. 1 is: "Typical Averaging Kernels (AKs, coloured solid lines) and vertical resolution (red dotted lines) of CCl4 VMR retrieved from Full Resolution (FR, top) and Optimized Resolution (OR, bottom) MIPAS measurements. The vertical resolution is calculated as the FWHM of the AK rows. The plot's key shows also the average number of degrees of freedom (DoF) of the retrieval (trace of the AK matrix) and the number of retrieval grid points (Npt)."

**p8 l 11 & 14. There are still quite a few minor English language problems in this manuscript, two examples here are "CCl4-poor" and "in the South Pole".**

The manuscript has been carefully proof read. We hope that the revised paper we are submitting to ACP is further improved.

**p8 l11. If there is a seasonal effect it is not obvious from figure 4. Can the authors quantify this seasonality, also to prove that it is indeed statistically significant? A similarly quantitative approach would help in other parts of the manuscript too, e.g. the earlier statements on latitudinal and altitudinal gradients.**

Sect. 3 has been modified. In particular we moved the comments on the seasonal variability to the description of Fig. 5. The seasonal variability of CCl4 distribution probably is not obvious from Fig. 4, however it is evident from Fig. 5.

**p9 figure 4 caption. "May 20117"**

Done. "May 20117" → "May 2011"

**p11 l9-11. This is not correct. Numerous aircraft and balloon campaigns have measured CCl4 with alternative in situ techniques. Please see e.g. Volk et al., 1997 and the many papers that cite it, as well as the FTIR total column measurements from the Jungfraujoch station.**

The sentence has been rephrased. The two instruments used in the paper for inter-comparison purpose are not the only ones available.

**p16 l4-6.This is exactly where alternative validation methods could help.**

We agree with the reviewer. In the revised paper we highlight that we do not pretend to carry-out a comprehensive validation work, we limit the intercomparison to MIPAS-balloon and ACE-FTS measurements.

**p23 l10. A "kind of global CCl4 trend"?**

Corrected. "A kind of" → "the"

**p24 l6. The smaller trend error does not take into account the biases, though.**

This is correct. The MIPAS finer sampling (with respect to ACE-FTS) permits to estimate trends with a smaller random error, i.e. with a better precision. The sentence has been rephrased. "With MIPAS it is therefore possible to achieve a smaller trend error." → "With MIPAS it is therefore possible to estimate trends with a better precision".

**p24 section 6. This section needs some additional work. The methodology (equation 2) is not used correctly as Plumb and Ko (1992) clearly state that a) it should only be applied to two species in steady state and b) the slope needs to be determined exactly at the tropopause. Moreover the method was improved by Volk et al.,1997 and Brown et al., 2013 to e.g. correct for tropospheric trends and derive steady-state lifetimes. A second problem with the lifetime estimate presented here is that it is highly dependent on uncertainties and potential biases of the trace gases involved, i.e. CCl4, CFC-11 and CFC-12. Can the authors present evidence that these uncertainties and biases have been taken into account for the determination of the lifetime and its uncertainties?**

Sect. 6 was re-written. CCl4 lifetime is now estimated using the method proposed by Volk et al. 1997 and Brown et al. 2013 that accounts also for the actual trend of the considered tracers. Since the actual trends of CCl4 and CFC-11 are rather small we get an estimate very similar to that presented in the discussion paper. To better characterize the uncertainty of our CCl4 lifetime estimate, we now include additional details on error calculation and also the results of some sensitivity tests we carried-out to evaluate the impact of some additional error components.

We would like to thank the reviewer for useful comments. In the following we answer the specific comments (included in "**boldface**" for clarity) and, whenever required, we describe the related changes implemented in the revised manuscript.

**Anonymous Referee #3**

**Overview**

**The paper presents the results of an analysis of the new MIPAS CCl4 product from the ESA processor. While opportunities for validation are limited, the authors do exploit one of the strengths of MIPAS, which is a 10-year globally sampled dataset to draw conclusions on interhemispheric variation and trends. On the whole, the paper is a clearly-written and convincing and I have no major criticisms.**

**General comments**

**a) While there is a convincing trend (matching the ground stations) it would have been useful to apply the same trend analysis to a different molecule retrieved with the same algorithm (eg N2O?) which has no expected trend. This would help quantify the contribution of any calibration drift.**

We have repeated the trend analysis for the N2O and, at least for pressures between 60 and 200 hPa we do not find any statistically significant trend at all latitudes. This finding confirms that the residual calibration drift error of MIPAS is very small, as already anticipated by the careful Level 1b studies carried-out by the MIPAS Quality Working Group team and already cited in the paper (see Sect. 2.1). In the revised paper, still we are not showing maps of N2O trends which are not a focus of the current study, we are considering N2O trends for a future additional publication.

**b) Of all the time-series fit parameters, it would have been helpful to indicate which ones were actually significant: the trend, constant and annual cycles are obvious from Fig 10 but what effect do the other terms have? Were they really needed?**

[Figure]

Figure 1A: Contribution of the different terms of the fitting function for 50°-55° S at 50 hPa (upper left panel), 50°-55° N at 50 hPa (upper right panel), 50°-55° S at 80 hPa (bottom left panel) and 50°-55° N at 80 hPa (bottom right panel). For each panel we report: in the first plot the fitted time series (FIT, red line) and the monthly zonal mean time-series (MZM, black line), in the second plot the residual time-series (RES, black line) calculated as MZM minus FIT; in the third plot the contribution of the sum of the periodicities (SUM, red line) and the MZM minus all the fitted terms excluding SUM (black line); in the fourth plot the contribution of the sum of the two QBO terms (QBO, red line) and the MZM minus all the fitted terms excluding QBO (black line); in the fifth plot the contribution of the solar radio flux (SRF, red line) and the MZM minus all the fitted terms excluding SRF (black line); in the sixth plot the contribution of the trend (TREND, red line) and the MZM minus all the fitted terms excluding TREND (black line).

In Figure 1A we show the contribution of the different terms of the fitting function for different pressure levels and different latitudes (see Fig 1A caption for more details). We can see that the amplitude of the contribution of the different terms of the fitting function depends both on latitude and pressure. In order to avoid discontinuities in the derived

trend values we decided to use the same fitting function (including all terms) for all the pressure / latitude bins, though for some of them, one or more terms of the fitting function may have small or negligible contributions.

**c) Comparison with ground stations: is the assumption here that the CCl4 profile is expected to be constant with altitude all the way through the troposphere? It would have been helpful to show at least a modelled CCl4 profile to support this. However, the fact that the MIPAS data have a seasonal cycle while the ground station data do not suggests that these must be different air masses, in which case there is presumably also some age difference between the air sampled by MIPAS and the surface air which could explain some of the bias.**

A thorough work on modeled CCl4 has been made by Chipperfield et al. (2016). The modeled CCl4 profiles shown in that paper are approximately constant in the troposphere. The comparison between CCl4 retrieved from MIPAS measurements and CCl4 model data is not a focus in this paper. This comparison will be the subject of a forthcoming work.

To highlight that the comparison is based on the hypothesis of well-mixed troposphere, we added the following sentence at the beginning of Sect. 5.3: "Under the assumption of well-mixed troposphere, we can consider the CCl4 vertical distribution approximately constant (Chipperfield et al., 2016; Allen et al., 2009)".

The new reference is:

Chipperfield, M. P., Liang, Q., Rigby, M., Hossaini, R., Montzka, S. A., Dhomse, S., Feng, W., Prinn, R. G., Weiss, R. F., Harth, C. M., Salameh, P. K., Mühle, J., O'Doherty, S., Young, D., Simmonds, P. G., Krummel, P. B., Fraser, P. J., Steele, L. P., Happell, J. D., Rhew, R. C., Butler, J., Yvon-Lewis, S. A., Hall, B., Nance, D., Moore, F., Miller, B. R., Elkins, J. W., Harrison, J. J., Boone, C. D., Atlas, E. L., and Mahieu, E.: Model sensitivity studies of the decrease in atmospheric carbon tetrachloride, Atmos. Chem. Phys., 16, 15741-15754, doi:10.5194/acp-16-15741-2016, 2016.

**d) Given the data available, it is possible to calculate a \*total\* atmospheric content of CCl4, at least the partial column above some pressure surface, and provide the trend of this with time. This would be a much easier quantity for simple**

**comparison with models or other satellite instruments without having to match details of pressure levels or latitude bands, also for stratospheric chlorine budgets.**

We used the approach presented in Sect. 5.1 to estimate also the trend of CCl4 partial column within two pre-defined pressure levels. For each monthly mean CCl4 profile referring to a latitude bin we calculated the partial column in the 10 - 100 hPa layer. For each latitude bin we then fitted the time-series of the partial columns using the fitting function (Eq. 1). We finally calculated the weighted average over latitude of the column trends, the weights being the cosine of the average latitude of the bin. For mean hemispheric trends we find $(-8.2 +/- 0.8) *10^{13}$ mol cm$^{-2}$ dec$^{-1}$ for SH and $(-12.3 +/- 0.8) * 10^{13}$ mol cm$^{-2}$ dec$^{-1}$ for NH. Dividing the monthly average columns in each latitude bin by the mission-average column of the same bin we also derive the following relative trends: $(-13.1 +/- 1.7)$ % dec$^{-1}$ for SH and $(-21.7 +/- 1.5)$ % dec$^{-1}$ for NH.

We decided not to include this exercise in the current paper due to the impossibility to make an exhaustive inter-comparison with other measurements. We have found only an atmospheric column trend estimation reported by Rinsland et al. (2012). They measured CCl4 atmospheric columns over Jungfraujoch (46.5 degN) finding a trend of $(-1.49 +/- 0.08) * 10^{13}$ mol cm$^{-2}$ yr$^{-1}$. In the 45/50 degN latitudinal band we found a trend of $(-1.15 +/- 0.08) * 10^{13}$ mol cm$^{-2}$ yr$^{-1}$. As mentioned earlier, the comparison of MIPAS measurements and CCl4 model data will be the subject of a forthcoming work and we would prefer to include the results of this exercise in that context.

**Minor comments**

**P2 L5: It is not clear from the text whether CCl4 is an entirely anthropogenic gas or whether there is also some (small?) natural source.**

The role of CCl4 natural sources is not completely clear and the magnitude of these natural emissions is not completely quantified. In the recent SPARC Report (2016) the authors indicate 3-4 Gg/year as the upper limit of the natural emissions.

To highlight this recent result, in the revised paper we added the following sentence in Sect. 1: CCl4 natural emissions are not completely understood and they are still under discussion. Stratospheric Processes and their Role in Climate (SPARC) community

(SPARC, 2016) recently defined an upper limit of the natural emissions (based on the analysis of old air in firn snow) of 3-4 Gg/year over a total emission estimation of 40 (25-55) Gg/year.

**P4 L19: If you mention 'oversampling the limb' you should explain what the size of the field-of-view is.**

MIPAS FOV is approximately 3 km in vertical. This information is now included in the mentioned paragraph.

**P4 L21: 8 rows for the FR AK, but only 7 for OR.**

We rephrased the sentence. This is consistent with the fact that the retrieval grid consists of 8 points (nodes) in the case of FR measurements and of 7 points in the case of OR measurements.

**P7 Much of the text here us unnecessary as it is already in the Fig 3 caption.**

Here we believe that the information reported in the text is important to understand the details of figure 3 and cannot be delegated uniquely to the figure caption.

**P9 Presumably the effect is larger in the antarctic due to the stronger, more stable polar vortex?**

OK. We included this comment in the revised paper.

**P10 L6: Since the ocean is the major surface sink, and there is more ocean in the southern hemisphere, wouldn't an IHG be expected even in the absence of continued emissions?**

If we compare CCl4 partial lifetime with respect to the ocean sink (209 years (Butler et al., 2016)) with the time needed by an air mass to move from the NH to the SH (around a year), we deduce that, in absence of emissions, the differences between NH and SH concentrations should be negligible. For a more rigorous explanation we refer to Liang et al. 2014.

**P11 L5/Fig 6: since Fig 6 is effectively an annual average its difficult to argue which components are persistent and which are seasonal. Perhaps there's an alternative**

**way of plotting the data to highlight the seasonal differences (eg shift the s. hemisphere data by 6 months before subtracting?)**

The figure was built without using a 6-months shift, but we have verified that a shift of 6 months does not change significantly the results since the impact of seasons is reduced by the average over a 7-years period. We revised the text of the paper explaining that the observed differences at high altitudes are not caused by the seasons but they are related to the asymmetry in the magnitude and in the persistence of the subsidence during winter and spring at the two poles.

**P11 L14: I can understand why balloon instruments might have better signal/noise than satellite instruments since they can effectively take many scans of the same atmosphere, but I don't understand what is intrinsic to the balloon measurement that gives it high vertical resolution compared to satellites. Indeed the 1.5km spacing of MIPAS-B seems comparable to MIPAS.**

We removed this sentence as it was not so important to understand the work presented in Section 4.1. The original intention was to explain that with a given angular aperture of the instrument FOV, the vertical resolution achieved from a stratospheric balloon platform is finer than that achieved from the satellite because the balloon is much closer to the sampled atmosphere. However MIPAS-B and MIPAS/ENVISAT instruments do not have the same angular FOV aperture.

**P15 L12: Given that CCl4 is a relatively long-lived gas with no diurnal variation, and that both MIPAS and ACE-FTS obtain relatively uniform sampling in longitude, I wonder why you didn't simply compare zonal means of both datasets (interpolating MIPAS to the appropriate latitude for ACE-FTS each day) rather than look for profile-by-profile coincidences which could contain a latitude bias or end up just selecting MIPAS ascending or descending node observations (with the associated GRAD error).**

As highlighted in the plot in the bottom panel of Fig. 3, in this part of the mission the GRAD error is expected to show a maximum value of only 3% at 15 km (approximately 120 hPa) and to rapidly decrease at higher altitudes. For this reason the GRAD error is not expected to play an important role in the inter-comparison with ACE. Moreover, since the horizontal resolution of MIPAS is at least as broad as 300 km for the weakest

species (see von Clarmann, T., De Clercq, C., Ridolfi, M., Höpfner, M., and Lambert, J.-C.: The horizontal resolution of MIPAS, Atmos. Meas. Tech., 2, 47-54, doi:10.5194/amt-2-47-2009, 2009) the matching criterion we use (300 km and 3 hrs) is quite stringent. Note that with our used matching method we also avoid the interpolation error that would be implied by the approach suggested by the reviewer.

**P15 L15: Again much of the text repeats what is in the figure caption, although it takes a while before explaining what I really wanted to know, which is the distinction between 'standard deviation of the mean' and 'standard deviation of the differences'. The former is just the latter divided by root(N), is that right?**

Yes, right. We modified the text to include this detail.

**P17 Eq(1): I agree with the approach but the term 'offset parameters' confused me - offset relative to what? Perhaps just 'constant parameters'.**

Done. We have replaced "offset parameters" with " constant parameters".

**Typographic/grammatical comments**

**P1 L1: no need for capital C in 'Carbon tetrachloride'**

Done.

**P1 L12: 20-50 rather than 20/50 if this indicates a range of latitudes rather than a particular pair of latitudes**

This sentence has been deleted.

**P3 L9: Similarly.**

Done.

**P2 L33: Suggest 'limits' rather than 'edges'.**

Done.

**P3 L14: 'where' rather than 'were'**

Done.

**P15 L6: Suggest 'extends' rather than 'goes'**

Done.

**Fig 5: some vertical lines at the year boundaries would be helpful.**

Done. We have modified Fig. 5 adding vertical dashed lines at the year boundaries. This information is now reported also in the caption.

**Fig 6: 'degN' for the latitude axis should presumably just be 'deg' here.**

Done.

We would like to thank the reviewer for useful comments. In the following we answer the specific comments (included in "**boldface**" for clarity) and, whenever required, we describe the related changes implemented in the revised manuscript.

**Anonymous Referee #4**

**I think that this is a useful and important paper which is well suited to publication in ACP. There has been a lot of interest in atmospheric CCl4 because of an apparent 'budget gap'. An important sink term for CCl4 is atmospheric loss and to evaluate our understanding of that process profile observations into the stratosphere are required. This paper presents such data from the MIPAS instrument which has the benefit of a lot of observations to average over.**

**I think that the paper can be published subject to my comments below.**

**Main points**

**1) Throughout the paper could benefit from a thorough proof-reading. There are some simple spelling errors that any spell checker should find. There are also some other sentences where the English is poor. The quality does vary through the paper (e.g. the abstract in particular had many typos). I have mentioned some below, but in addition the paper needs careful proof reading.**

Probably the reviewer refers to the initially submitted version of the paper. The version published in ACPD was carefully proof read and the language was also revised. We hope that the revised paper we are submitting to ACP is further improved.

**2) Stratospheric trends. A number of recent papers have shown that the trends in stratospheric trace gases are affected by variability in the stratospheric circulation. This has been shown for a number of halogen source gases and the complementary degradation products such as HCl and HF. This is bound to be playing a role in the stratospheric trends shown in Figure 11 and will be at least part of the explanation of why the trend does not simply follow the tropospheric trend (with a lag). I know there is mention in the Conclusions (page 26 line 5) but more should be added near Figure 11. It is a case of adding in some mention of past work. Examples to cite are:**

Harrison, J.J., M.P. Chipperfield, C.D. Boone, S.S. Dhomse, P.F. Bernath, L. Froidevaux, J. Anderson and J.M. Russell, Satellite observations of stratospheric hydrogen fluoride and comparisons with SLIMCAT calculations, Atmos. Chem. Phys., 16, 10,501-10,519, doi:10.5194/acp-16-710501-2016, 2016.

Mahieu, E., M.P. Chipperfield, J. Notholt, T. Reddmann, J. Anderson, P.F. Bernath, T. Blumenstock, M.T. Coffey, S. Dhomse, W. Feng, B. Franco, L. Froidevaux, D.W.T. Griffith, J. Hannigan, F. Hase, R. Hossaini, N.B. Jones, I. Morino, I. Murata, H. Nakajima, M. Palm, C. Paton-Walsh, J.M. Russell, M. Schneider, C. Servais, D. Smale and K.A. Walker, Recent northern hemisphere hydrogen chloride increase due to atmospheric circulation change, Nature, 515, 104-107, doi:10.1038/nature13857, 2014.

Ploeger, F., Riese, M., Haenel, F., Konopka, P., Müller, R., and Stiller, G.: Variability of stratospheric mean age of air and of the local effects of residual circulation and eddy mixing, J. Geophys. Res.-Atmos., 120, 716–733, doi:10.1002/2014JD022468, 2015.

We thank the reviewer for the useful comment. We partially included the above sentences in Sect. 5.2: "Recently some studies (Harrison et al., 2016; Mahieu et al., 2014; Ploeger et al., 2015) have shown that the trends in stratospheric trace gases are affected by variability in the stratospheric circulation. This has been shown for a number of halogen source gases and the complementary degradation products (i.e. HCl and HF). This variability can partially explain why the stratospheric trend does not simply follow the tropospheric trend with a lag." The references to the three suggested papers are now included in the revised paper.

**3) Figure 6 does not make sense to me. Normally the N-S IHG is presented based on an average over the two hemispheres. How is Figure 6 constructed? Is it the difference between corresponding latitudes (e.g. 80S minus 80N)? That does not make sense as the high latitudes get more and more distant from the other hemisphere so the scope for differences is much larger. There is also less mass at high latitudes so the differences are not so important in a budget sense. I think that this figure is flawed and should be removed.**

Figure 6 is constructed as a mean on seven years of the differences between CCl4 VMR profiles in the Northern Hemisphere (NH) and Southern Hemisphere (SH) at corresponding latitudes. The large differences at high latitudes are due to the fact that the subsidence of air in the SH has a longer duration than in the NH. Generally subsidence occurs until November in the SH, but only until March in the NH. Usually the North-South IHG is defined as single number representing the difference between the average VMR in the two hemispheres. In the case of MIPAS, however, we have the great opportunity to compute the temporal average of the North-South VMR differences for each pressure level and latitude bin. This is why we would prefer to keep Fig. 6, although we agree that its description should be improved.

In order to compare our results with the North-South IHG reported in the literature (Liang et al., 2014), in the revised paper we compute also the latitudinal-average of the NH-SH VMR differences. For each pressure level this is obtained by weighting the monthly mean VMR in a given latitude bin with its corresponding solid angle fraction. These results are now discussed in the revised paper.

**Minor Points**

**Abstract line 1. Change 'strong' to 'potent'?**

Looking in the web, the construction "strong ozone-depleting substance" seems more popular than "potent ozone-depleting substance".

**Page 1. Line 4. Typo: mystery**

Already done in the last version of the discussion paper.

**Page 1. Line 6. Typo: photolytic**

Already done in the last version of the discussion paper.

**Page 1. Line 9. Typo: anthropogenic**

Already done in the last version of the discussion paper.

**Page 1. Line 12. 'proves' is too strong. Could change to 'gives confidence in' (or similar).**

The sentence has been rewritten.

**Page 1. Line 16. Change scan to scans.**

Already done in the last version of the discussion paper.

**Page 2. Line 1. ODP is ozone \*depletion\* potential.**

Done.

**Page 2. Line 6. Typo: hydrofluorocarbons**

Already done in the last version of the discussion paper.

**Page 3. Line 4. Typo: where**

Done.

**Page 3. Line 5. Here you could cite a recent paper on modelling the CCl4 budget using the latest lifetime data and limited ACE CCl4 data to evaluate the model stratosphere. The availability of more stratospheric data would help constrain such model studies.**

**Chipperfield, M.P., Q. Liang, M. Rigby, R. Hossaini, S.A. Montzka, S. Dhomse, W. Feng, R.G. Prinn, R.F. Weiss, C.M. Harth, P.K. Salameh, J. Muhle, S. O'Doherty, D. Young, P.G. Simmonds, P.B. Krummel, P.J. Fraser, L.P. Steele, J.D. Happell, R.C. Rhew, J. Butler, S.A. Yvon-Lewis, B. Hall, D. Nance, F. Moore, B.R. Miller, J.W. Elkins, J.J. Harrison, C.D. Boone17, E.L. Atlas and E. Mahieu, Model sensitivity studies of the decrease in atmospheric carbon tetrachloride, Atmos. Chem. Phys., 16, 15,741-15,754, doi:10.5194/acp-16-15741-2016, 2016**

In the revised version of the paper we now cite also the above mentioned paper.

**Page 3. Line 21. 'operation' (singular).**

Done.

**Page 3. Line 32. Change to 'allowing the study of the evolution of atmospheric**

**composition in great detail'.**

Done.

**Page 4. Table 1. Spell out MW in the caption.**

Done.

**Page 4. Line 12. Change to 'includes only one out of every two'.**

Done.

**Page 5. Figure 1 caption. Specificy 'coloured solid lines'.**

Done.

**Page 5. Line 4. 'Apart from the "NLGAIN"...'**

Already done in the last version of the discussion paper.

**Page 6. Line 7. Do these errors 'cancel out' exactly? If not you should say something like 'largely cancel out. . .'.**

Done.

**Page 7. Line 2. Typos: '. . ..type of error, therefore, has no impact on the trend calculation'.**

Done.

**Page 7. Line 19. 'We do not show..'**

Done.

**Page 7. Lines 25-29. These lines are not clear to me. I think it is the use of the word 'compatible'. You should look into rephrasing this.**

Here we mean "compatible" from the statistical point of view. This terminology seems quite common in error analysis discussions.

**Page 8. Line 5. 'continuing for inertia'. This does not make sense and needs to be rephrased.**

The section has already been rephrased in the last version of the discussion paper.

**Page 8. Line 11. 'hemispheres' (small h).**

Done.

**Page 8. Line 12. 'troposphere' must be a typo? At 130 hPa high latitudes will be in the stratosphere.**

Done. "in the troposphere" ➔ "at the lowermost pressure levels".

**Page 8. Line 14. Change 'notice' to 'note'.**

Done.

**Page 9. Line 7. Typo: 'transport'.**

Already done in the last version of the discussion paper.

**Page 9. Line 8. 'justify' is the wrong word. Use 'explain'?**

Already done in the last version of the discussion paper.

**Page 12. Line 7. 'Further to'. 'simultaneously'.**

Done.

**Page 12. Line 20. 'incompatible'.**

Done.

**Page 13. Figure 7 (and 8). The caption should explain the red numbers on the left panel.**

The captions of Fig. 7 and Fig. 8 have been modified taking also into account this comment.

[revised manuscript text omitted]
 seasonal variability is also clearly visible from the maps of Fig. 4, with opposite phases in the two Hemispheres, more pronounced at mid-latitudes and in the polar regions. As previously mentioned, in the troposphere a minimum is found in the Northern and Southern Polar Spring.gradientnotice~~ note that for pressures larger than 100 hPa, the $CCl_4$ measured in the OR phase has a positive bias with respect to that measured in the FR phase. This bias, discussed also in Sect. 4.1, may be due to the different MWs used for the retrieval in the two mission phases, or to the different limb sampling patterns adopted.

 The Inter Hemispheric Gradient (IHG) at the surface is  largely used as a qualitative indicator of  continuous emissions (Lovelock et al., 1973; Liang et al., 2014). Anthropogenic emissions are larger in the  Northern Hemisphere (NH) (SPARC, 2016) and the transport of these emissions from the NH to the  or more. Southern Hemisphere (SH) takes about one year, i.e. a time interval much shorter than the $CCl_4$ lifetime (see Sect. 6). Hence, a significant IHG  driving the hemispheric differences. At higher altitudes, the asymmetry between the North and South  $CCl_4$  distribution represents evidence of ongoing emissions.

Although MIPAS measurements are not suitable to evaluate the IHG at the surface , they provide information about the distribution of inter-hemispheric differences in the UTLS region as a function of both latitude and pressure. To analyze these differences we interpolated to a fixed pressure grid MIPAS $CCl_4$ profiles acquired from April 2005 to March 2012. We then binned the profiles in 5° latitude intervals and calculated, for each latitude bin, the  average $CCl_4$ VMR profile in the considered time period. Finally, for each latitude bin in the NH we identified the corresponding bin in the SH and computed the difference between the average profiles. The  Fig. 6  in the stratosphere originate from the subsidence effect during polar winter and spring, bringing mesospheric -poor air in the stratosphere. This effect is generally larger in the SH 
[revised manuscript text omitted]